# Molecular signatures and causal factors underlying latent cytomegalovirus infection among people living with HIV (PLHIV)

Nhan Nguyen [1,2,9], Zhenhua Zhang [1,2,3,9], Xun Jiang [1,2,9], Nienke van Unen [1,2,9], Jéssica C. dos Santos[4], Liang Zhou [1,2], Vasiliki Matzaraki [4], Javier Botey-Bataller [1,2,4], Marc Blaauw[4], Wilhelm A. J. W. Vos[4,5], Louise van Eekeren[4], Annelies Verbon [6,7], Albert L. Groenendijk [4,6,7], Janneke E. Stalenhoef [5], Marvin A. H. Berrevoets[8], Leo A. B. Joosten [4], Mihai G. Netea [4], Cheng-Jian Xu[1,2], André J. A. M. van der Ven [4] & Yang Li [1,2,4] ✉

CMV seropositivity contributes to medical complications in people living with HIV (PLHIV). This study provides a comprehensive evaluation of how CMV seropositivity shapes the immune system of 1,887 PLHIV, by utilizing multi-omics and deep immune phenotyping datasets. The study measured the immune cell profiles from whole blood, and the cytokine production of PBMCs exposed to various ex-vivo stimuli. We observed an increase in pro-inflammatory cytokine production of circulating immune cells and differences in phenotype of innate-like lymphocyte populations associated with CMV seropositivity. This study also measured 5-omics layers, including genomics, DNA methylation, transcriptomics, and plasma protein and metabolites. The DNA methylome and transcriptome demonstrated prominent CMV-induced signatures related to immune functions in PLHIV. Particularly, high *FCRL6* expression is a promising biomarker for immune activation, underlined by the demethylation of *FCRL6* and up-regulation of gene expression and plasma protein concentrations in CMV-seropositive PLHIV. Host genetics-driven elevation in both gene and protein expression of FCRL6 was also associated with latent CMV infection. A significant CMV-seroprevalence locus was associated with cytokine production capacity and protein abundance. Mendelian randomization analyses demonstrated a causal relationship between elevated *FCRL6* expression and CMV seropositivity.

Cytomegalovirus (CMV), an ubiquitous member of the *Herpesviridae* family, is a double-stranded DNA virus with a ~236 kB genome. After primary infection, CMV establishes lifelong latency in early myeloid cells, including CD34+ haematopoietic stem cells and monocyte progenitor cells, as well as in blood monocytes[1] and possibly also in other immune cells. CMV can reactivate when host immunity wanes, which threatens pregnant women (leading to congenital CMV) and immunocompromised patients. Estimated CMV seroprevalence ranges between 50 and 90% of the global population, depending on geography, behavior, and co-infection patterns[2,3].

Before combination antiretroviral therapy (cART) became standard for people living with HIV (PLHIV), CMV disease, such as retinitis,

---

colitis, pneumonitis, and encephalitis, occurred in up to 40% of patients with advanced HIV infection[4]. Modern cART suppresses HIV replication and partly restores immunity, yet many virally suppressed PLHIV still exhibit chronic immune activation, accelerated immune ageing, and higher rates of non-AIDS comorbidities[5]. Hence, asymptomatic virally-suppressed PLHIV using ART may still face some clinical consequences of a CMV co-infection, as recently reviewed by Freeman[6].

Previous studies have investigated the effects of CMV infection, with a primary focus on specific biological processes such as DNA methylation[7], gene expression[8], or metabolism[9]. However, these single-layer studies cannot provide a comprehensive understanding of molecular signatures and driving factors of CMV infection. Such a systems-level picture is particularly important for PLHIV, who show both a very high CMV prevalence and persistent immune dysfunction despite suppressive cART[5].

Thus, the 2000HIV study[10], including 1895 HIV-suppressed PLHIV from the Netherlands, aims to unravel biological processes associated with HIV clinical phenotypes using functional immunological assays, de*f*ep immune phenotyping, and multi-omics analysis. Around 6% of 2000HIV study participants were CMV-seronegative (CMV-), allowing the opportunity to comprehensively investigate how latent CMV may shape the immune system of PLHIV.

Here, we pinpoint the molecular signatures that distinguish CMV-seropositive (CMV+) from CMV- PLHIV. We first show that CMV seropositivity associates with enhanced cytokine-production capacity and altered immune-cell composition. At the molecular level, CMV seropositivity is associated with widespread DNA methylation alterations that lead to substantial changes in gene expression and affect immune-related processes. Across layers, the Fc receptor-like gene (*FCRL6*) emerges as a consistent marker of CMV co-infection in PLHIV. It is demethylated in whole blood cells of CMV + PLHIV, leading to upregulation at both gene expression in circulating immune cells and plasma protein abundance levels. Furthermore, Mendelian Randomization (MR) analyses reveal that genetic variants increasing FCRL6 expression or protein levels may causally raise CMV seropositivity risk. Complementary genome-wide association analysis uncovers a separate chromosome-15 locus linked to CMV status, cytokine production, and NK-cell receptor abundance. Together, these data provide the first multi-omics, functional, and genetic map of CMV co-infection in a large cohort of treated PLHIV, laying the groundwork for therapeutic strategies aimed at mitigating CMV-driven immune activation in HIV infection (Supplementary Fig. S1).

## Results

### A large PLHIV cohort with comprehensive multi-omics and immune profiling data

People living with HIV (PLHIV) on antiretroviral therapy (ART) may still experience long-term immune dysregulation. To dissect the underlying mechanisms, the 2000HIV study adopted a systems-immunology approach that combines multi-omics with functional immune assays. The study comprises a discovery (*n* = 1553) cohort across three Dutch HIV treatment centers and an entirely independent validation cohort (*n* = 334) from a fourth center, enabling immediate replication of key findings in an independent sample set (Fig. 1A)[10]. The combined cohort was predominantly male (84.7%) with a mean age of 51.6 years. Baseline cytomegalovirus (CMV) IgG was measured in all 1887 participants by ELISA, revealing seronegativity in around 6% of the discovery cohort (*n* = 86) and around 8% of the validation cohort (*n* = 28, Supplementary Data S1).

The resulting dataset spans genome-wide single-nucleotide polymorphisms (SNPs), whole-blood DNA-methylation profiles, bulk transcriptomes generated from peripheral blood mononuclear cells (PBMCs), targeted plasma proteomes, and untargeted plasma metabolomes. Whole-blood, high-dimensional flow-cytometry immunophenotyping provided cell counts of major innate and adaptive immune-cell subsets, while ex vivo stimulation of PBMCs with innate and adaptive ligands followed by cytokine quantification at 24 h and 7 days captured functional immune capacity (Fig. 1A). Comparative analyses in the discovery cohort uncovered significant CMV-associated differences across multiple omics layers and immune-phenotyping readouts (Fig. 1B, Supplementary Data S2–S12), suggesting that latent CMV infection imprints a distinct molecular and functional signature on the immune system of PLHIV. The present work, therefore, focuses on defining the core genomic, epigenomic, transcriptomic, cellular, and cytokine-response features that characterize CMV latency within chronic, treated HIV infection.

### Latent CMV infection heightens pro-inflammatory recall responses in PLHIV

Among the 90 cytokine-stimulus read-outs measured, the most pronounced CMV-associated effects appeared after exposing PBMCs to the CMV antigen pp65 for 24 h (Fig. 1C, Supplementary Data S2–S3). In both the discovery and validation cohorts, CMV-seropositive (CMV+) donors secreted significantly higher amounts of the monocyte-derived cytokines IL-1β, IL-1Ra, IL-8, and MCP1 than CMV-seronegative (CMV-) donors (FDR < 0.05 discovery; *P* < 0.05 validation). This agrees with previous studies where heightened inflammatory responses in PBMCs to CMV infection for PLHIV are already well-documented[11–13].

As pp65 is an internal CMV protein that does not directly engage pattern-recognition receptors, a possible explanation for the elevated innate-cytokine output is that the antigen is processed and presented to pre-existing pp65-specific T cells. Activated CD8 + T cells are known to release TNF-α and IFN-γ upon pp65 recognition, a phenomenon termed memory inflation, and the resulting IFN-γ allows monocytes to amplify IL-1β, IL-1Ra, IL-8, and MCP-1 production[14,15]. Consistent with this prior knowledge, whole-blood immunophenotyping revealed higher frequencies of HLA-DR + CD4+ and CD8 + T cells, NK cells, and gamma-delta (γδ) T cells in CMV + PLHIV (Fig. 1E; Supplementary Data S4, S5). This mirrors the well-documented CMV-driven reshaping of CD4 + , CD8 + , NK-cell, and γδ-T-cell compartments in healthy adults[15,16] and the capacity of CMV to upregulate HLA-DR expression on T cells[17]. Together, these findings establish that latent CMV infection primes a T-cell-driven, monocyte-dominated inflammatory response that distinguishes CMV+ from CMV- PLHIV.

### Multi-layer molecular overview of CMV seropositivity in PLHIV

Across multi-omics levels, we observed substantial differences in molecular signatures associated with CMV seropositivity (Fig. 1B, Supplementary Data S6–S12). In particular, >16,000 CpG sites showed significant DNA methylation differences associated with the CMV seropositivity in PLHIV (FDR < 0.05). A previous study has also highlighted that CMV infection elicits extensive changes in the DNA methylome in the blood of healthy adults[18]. Along with the steady changes in methylome, the expression of 1442 genes was significantly associated with CMV seropositivity in PLHIV (FDR < 0.05), using bulk PBMC transcriptomics data. By contrast, only 5 out of 1720 plasma metabolites were significantly associated with CMV seropositivity in PLHIV at the discovery cohort (FDR < 0.05, Supplementary Data S11), even though some studies found changes in metabolic pathways related to CMV infection[19]. Furthermore, these metabolites did not show statistically significant differences associated with CMV seropositivity in the validation cohort (Supplementary Data S12). Out of 2367 measured plasma proteins, the abundance of 38 proteins was significantly associated with the CMV seropositivity in the discovery cohort (Supplementary Data S9). Notably, the abundance of 18 out of these 38 proteins has shown similar differences in the validation cohort (Supplementary Data S10). The inflammatory markers include granzymes (GZM), Killer cell lectin-like receptor D1 (KLRD1), Adhesion G protein-coupled receptor G1 (ADRG1), and Fc receptor-like 6 (FCRL6), which

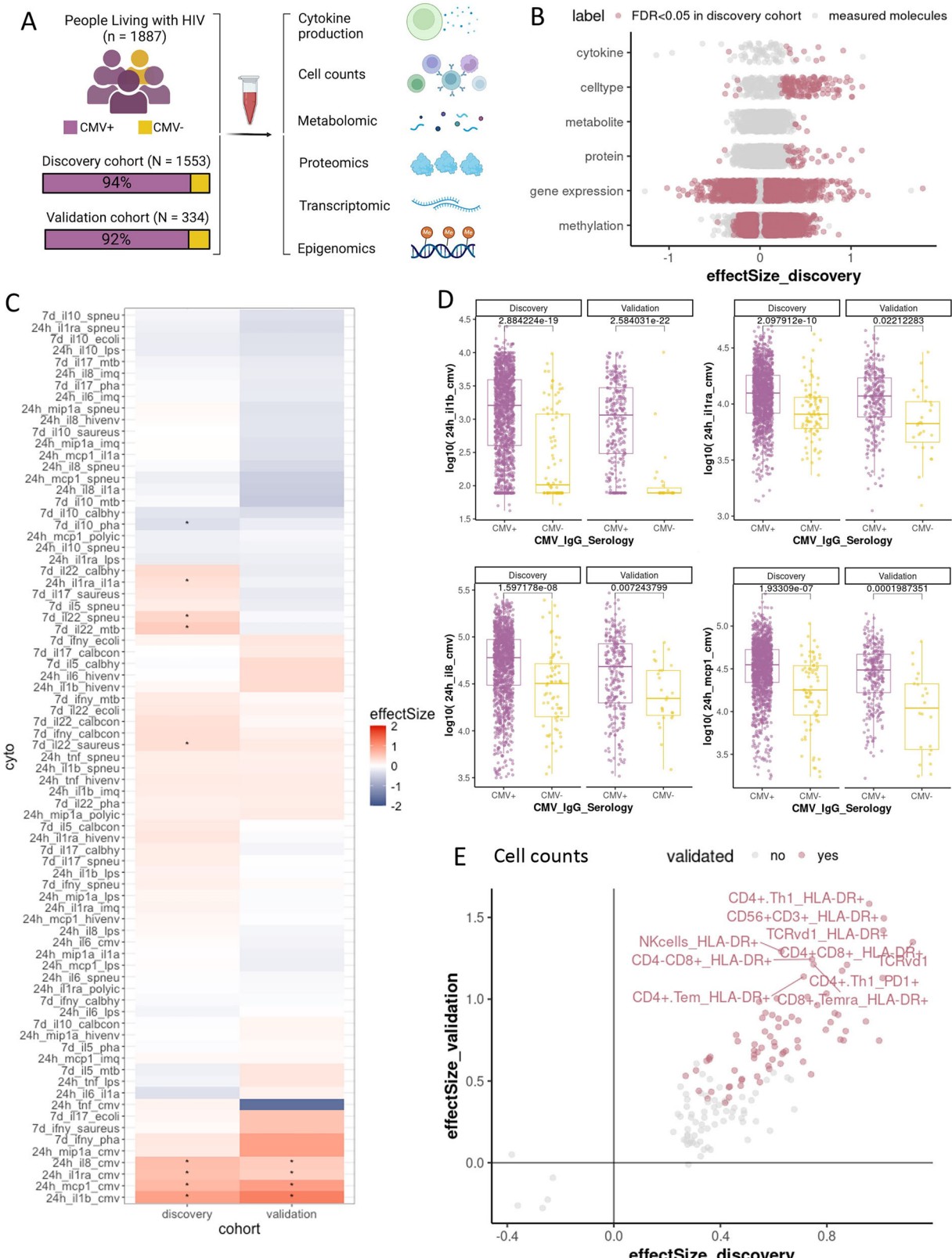

are produced by NK cells, γδ T cells, and CD8 + T cells[20], suggesting increased activation of both innate and adaptive immune cells in the context of CMV-HIV co-infections. Together with the concordant gene expression shifts and the underlying DNA-methylation differences that likely drive them, this protein signature supports a model in which latent CMV amplifies systemic inflammation in treated HIV infection, a mechanism that may contribute to the higher burden of CMV-related complications observed in this population. The following sections dissect these three layers in turn, beginning with DNA methylation, then the transcriptome and proteome.

**Fig. 1 | Cohort, multi-omic profiling, and immunological signatures. A** 2000HIV cohort with CMV serology information. Created in BioRender. Gödecke, S. (2025) https://BioRender.com/dz2rpem. **B** Multi-omics signatures were significantly associated with CMV seropositivity in the discovery cohort. **C** Cytokine production difference between CMV+ and CMV- individuals in discovery and validation cohorts after correcting for confounders. The cytokines were clustered based on the similarity in their sample profiles to each other. The red color means the cytokine production increased in CMV + PLHIV compared to CMV- PLHIV. The blue color means the cytokine production decreased in CMV + PLHIV compared to CMV-PLHIV. The cytokine production significantly associated with CMV serology status

(FDR < 0.05) is labeled with (*). **D** Boxplot of validated cytokine production significantly associated with CMV seropositivity in discovery and validation cohorts. In boxplots, center lines represent medians, boxes indicate 25th and 75th percentiles, and whiskers expand to 1.5× the interquartile range. **E** The absolute cell count of immune cell subsets which were associated with CMV seropositivity. The dots present cell subsets significantly associated with CMV seropositivity in discovery cohorts (FDR < 0.05). The purple color presents cell subsets which were validated in the validation cohort (P < 0.05). All statistical tests were conducted using robust linear models adjusted for confounding effects, with two-sided testing and multiple-test adjustments by FDR.

## CMV seropositivity is strongly associated with DNA methylation changes

To identify the epigenetic modifications associated with CMV infection, we performed epigenome-wide association studies (EWAS) of the whole blood DNA methylation data in both discovery and validation cohorts. We observed massive DNA methylation changes related to CMV seropositivity status across the entire genome; the nonparametric multidimensional scaling approach (NMDS)[21] illustrates the significant methylation difference associated with CMV infection (Fig. 2A, adonis, R2 = 0.005, P = 0.0005, with 1999 permutations). In the discovery cohort, we identified 16,245 differentially methylated CpG sites (DMSs) from 793,767 CpG sites (Fig. 2B; Supplementary Fig. S2A-C; FDR < 0.05) associated with CMV seropositivity in PLHIV, of which 14,536 DMSs were validated in the validation cohort (Fig. 2C), and more than 85% could be replicated in the downsampled analysis (Supplementary Fig. S3). Notably, 98.7% of the validated DMSs can be replicated in EWAS based on data generated from people without HIV[18] (Fig. 2D), and 78.6% of the significant CMV-associated DMSs detected from a large healthy French cohort (23,616 DMSs with FDR < 0.05) could be replicated in the 2000HIV discovery cohort (Fig. 2E). This finding suggests a high degree of consistency in DNA methylation differences caused by CMV seropositivity in PLHIV and healthy individuals.

To understand the impact of CMV-induced epigenetic modification, we explored potential TF motifs near validated DMSs using HOMER[22,23] motif enrichment analysis. We identified significantly enriched TF motifs, including FOS[24], FRA1[24,25], BATF[26], and ATF3[27] (Fig. 2F), which play roles in response to stress signals, apoptosis, inflammation, immune modulation, cell proliferation, and cell differentiation. From the validated DMSs, we diagnosed cis-expression Quantitative Trait Methylation (cis-eQTM) significant genes (Method, Supplementary Data S6, 4333 genes, FDR < 0.05) and performed functional enrichment analysis with the GREAT database[28] (Supplementary Fig. S2D). Thereby, we recognized that genes related to CMV-associated DMSs are significantly enriched in crucial biological processes for immune function and disease pathogenesis, such as mononuclear cell differentiation, T-cell differentiation and activation, B-cell activation, extracellular matrix organization, and cell motility (Fig. 2G, Supplementary Fig. S2D, E; FDR < 0.05).

To investigate the interaction between validated DMSs and immune response changes associated with CMV seropositivity, we conducted mediation tests[29] between CMV serostatus and the CMV-associated cytokines from PBMCs (Fig. 1D; Methods). Mediation effects from many DMSs were detected, which suggests that DNA methylation may partially contribute to the elevated innate immune responses (IL1β, IL1Ra, and IL8 production capacity) upon CMV stimulation, and make a direct contribution to adaptive immune response (IL-1ra, IL-10, and IL-22). Such effects may be conducted by the DMS-influenced genes (Fig. 2G). Among these DMSs, cg15843262 was associated with DNMT3A[30,31] (Fig. 2H, I; Supplementary Fig. S2F) and can mediate approximately 23% of IL1Ra response to CMV stimulation (Fig. 2J, upper panel). Further mediation analysis suggests that cg15843262 mediated CMV's impact on DNMT3A expression (Fig. 2J,

lower panel). DNMT3A encodes DNA methyltransferase 3A, which plays a pivotal role in de novo DNA methylation during development. The CMV-associated DMS in the DMNT3A region could alter the transcription of DMNT3A (Fig. 2H) and thus potentially elucidate the widespread DNA methylation changes associated with CMV seropositivity (Fig. 2A). Hence, latent CMV infection induces DNA methylation alterations that influence immune-related processes and mediate immune response changes.

## CMV-related molecular signatures at the transcriptome and proteome levels

We analyzed the bulk RNA sequencing data of 1496 PBMC samples in the discovery cohort and identified 1442 genes significantly associated with the CMV serostatus in PLHIV (DEGs, FDR < 0.05, Supplementary Data S6). Among them, 700 DEGs (48.54%, FDR < 0.05) were validated in the validation cohort (Supplementary Data S7). We performed pathway analysis in 807 DEGs validated in the validation cohort (P < 0.05), including 390 up-regulated and 417 down-regulated DEGs (Fig. 3A), resulting in 155 and 11 enriched pathways, respectively (Fig. 3B, Supplementary Data S8). A downsampling analysis was performed 100 times with random sample selections to evaluate the potential bias due to the imbalance between the CMV+and CMV-patient group size. On average, around 71.49% of 1442 DEGs in the discovery cohort were replicated in the downsampling analysis (FDR < 0.05) (Supplementary Fig. S4). Upregulated DEGs are involved in various immune processes such as different NK cell functions, cytotoxicity, innate defense response, signaling, and regulation pathways (Fig. 3C, D). Cell-mediated immunity plays a crucial role in CMV control, especially CD8 + T cells, CD4 + T cells, NK cells, and γδ T cells[32,33]. The increased immune function found in gene expression data of PBMCs (Fig. 3D) is in line with the elevated proportion of HLA-DR+ immune cell subsets (Fig. 1E). While CMV-specific cell-mediated immune assays via interferon gamma signaling can be used to predict the development of clinical manifestations of CMV or identify transplant recipients with low risk of CMV infection treatment after transplantation[34–36], other significant pathways arise from transcriptomic analysis, such as phosphatidylinositol 3 kinase signaling or CD4-positivity, alpha-beta T cell lineage commitment, may be used to further support the prediction tests. Our finding of upregulated DEGs in innate and adaptive immune responses (Fig. 3D), as well as the increased expression of plasma inflammatory proteins in CMV seropositive PLHIV, suggests that CMV seropositivity has a significant impact in individuals with a dysfunctional immune system[33]. After the infection phase, CMV may continuously impact both innate and adaptive immunity during persistent and latent phases[37].

We analyzed the plasma proteome of 1523 samples in the discovery cohort and detected the protein abundances of 38 proteins that were significantly associated with the CMV seropositivity in PLHIVs (DEPs, FDR < 0.05, Supplementary Data S9). The 18 proteins among those differential abundance proteins (47.37%) were up-regulated in CMV+ compared to CMV- PLHIV, and were validated in the validation cohort (P < 0.05) (Fig. 3E, Supplementary Data S10). To evaluate the potential bias due to the imbalance between the CMV+and CMV-

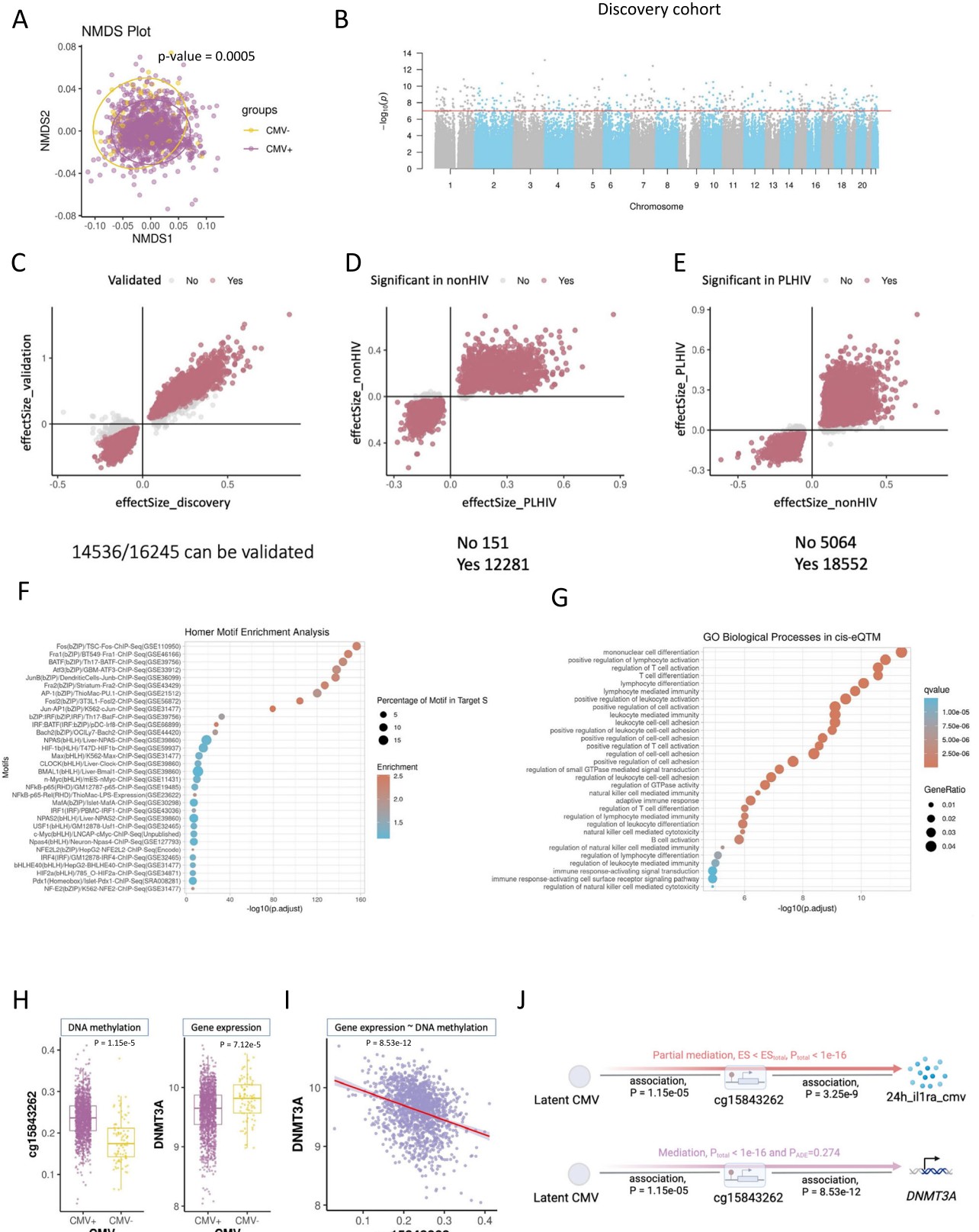

patient group size, we performed a downsampling analysis with random sample selections 100 times. On average, around 80.13% significant proteins in the discovery cohort were replicated in the downsampling analysis (FDR < 0.05) (Supplementary Fig. S5). Importantly, eight validated proteins were also significantly upregulated at gene expression levels in CMV + PLHIV across both discovery and validation cohorts and are also recognizable inflammatory markers (Fig. 3F; Supplementary Fig. S6A–C). Thus, CMV infection may not only impact gene expression of circulating immune cells but also influence systemic inflammation status.

To determine whether the CMV-associated proteomic changes we observed in PLHIV are HIV specific, we analysed the 300BCG cohort,

**Fig. 2 | CMV-associated signatures, methylation analysis. A** NMDS analysis between CMV+ and CMV-. **B** Manhattan plot of the EWAS results in different studies. -log10(P) of all the detected CpGs (*y*-axis) were plotted with the location (*x*-axis) through the genome. BACON adjustment was performed to suppress the inflation. The red line stands for 1e-7. **C** Scatter plot of the identified DMSs. *x*-axis: effect size extracted from the EWAS based on the discovery cohort. *y*-axis: effect size extracted from the EWAS based on the validation cohort. Red dots are the validated DMSs. FDR_dis<0.05, FDR_val<0.05, the effect size in the same direction. **D** Scatter plot of the effect size of the validated CMV-induced DMSs in PLHIV vs. non-HIV people. **E** Scatter plot of the effect size of the Bofferoni significant DMSs in non-HIV people vs. PLHIV. **F** HOMER motif enrichment analysis. 100 bp up- and downstream of the DMSs. **G** Functional enrichment analysis (GO, KEGG, and MF) of the cis-eQTM identified genes (FDR < 0.05, 250k up- and downstream of the CpG site). The top five of each were plotted. **H** Methylation of cg15843262 and gene expression of DNMT3A between CMV- and CMV+. *n* = 1512. Boxplots show the median (center line), 25th–75th percentiles (box), and whiskers extending to 1.5× the interquartile range. Individual samples are shown as jittered dots. Statistical tests were conducted using robust linear models adjusted for confounding effects, with two-sided testing and multiple-test adjustments by FDR. **I** The correlation between Methylation of cg15843262 and gene expression of DNMT3A between CMV- and CMV+. Statistical tests were conducted using robust linear models. **J** Summary of the association and mediation between CMV, cg15843262, DNMT3A, and cytokine response. Created in BioRender. Jiang, X. (2026) https://BioRender.com/e1qeosx.

---

including CMV+ (*n* = 74) and CMV- (*n* = 241) healthy individuals, which measured 73 proteins using an inflammatory Olink panel[38]. Nine proteins out of the 273 significant proteins (*P* < 0.05) were shared between datasets, of which three were nominally significant in both cohorts and all with concordant direction of effect (Supplementary Fig. S6D). Though the sample size in the healthy cohort is relatively small, this result indicates that the CMV-driven protein alterations appear to be independent of HIV status.

### CMV-responsive signals converge on cardiopulmonary and HIV-progression factors

Building on our previously published Multi-omics Factor Integration (MOFA) model of the European discovery subset (*n* = 996)[39], we asked whether CMV-associated molecular traits map onto the latent pathways already linked to comorbidities in PLHIV. We tested for over-representation of the CMV+ and CMV- differentially expressed molecules among the highest-weight features of the four disease-associated latent factors (LF6 plaque, LF8 hypertension & myocardial infarction, LF11 COPD, LF20 rapid progressors). We excluded DNA methylation from this analysis due to the direction of effect being more ambiguous than the other layers. Loadings were classified as risk or protective according to the direction of each factor's correlation with its disease phenotype.

Three of the four clinical latent factors (LF8, LF11, and LF20) show a clear CMV imprint (FDR < 0.05 across several top-weight cut-offs; Supplementary Fig. S7). LF11 is particularly striking. This factor, originally linked to both COPD risk and higher "latest CD8" counts, shows a significant positive correlation with CMV serostatus and a linear relationship with circulating CMV-IgG titres (Supplementary Fig. S7A, B). Its risk weights are enriched for CMV-upregulated genes and plasma proteins, whereas the protective tail is over-represented by CMV-down-regulated transcripts (Supplementary Fig. S7C). This polarity suggests that latent CMV infection drives transcriptional programs that feed into cytotoxic profiles captured by LF11, thereby predisposing to COPD pathogenesis. By contrast, transcripts reserved in CMV- individuals may buffer against excessive CD8+ activation and lung inflammation. This is not surprising, as COPD pathogenesis is tightly coupled to excessive CD8+T-cell-driven inflammation[40]. In addition, while the impact of CMV co-infection on COPD has not yet been explored in PLHIV, chronic CMV infection itself has been associated with COPD in HIV-uninfected populations[41].

For LF8, the factor linked to cardiovascular diseases, the risk features were enriched for CMV-upregulated protein expression (Supplementary Fig. S7D). This agrees with earlier reports linking CMV co-infection to cardiovascular comorbidity in PLHIV[5]. LF20, capturing rapid-progressor HIV phenotype, showed a similar CMV+ enrichment in its risk weights for gene expression signatures (Supplementary Fig. S7E), echoing evidence that CMV accelerates HIV disease progression[42]. All in all, the patterns shown here suggest that CMV infection amplifies molecular programs promoting heart and lung inflammatory diseases and HIV disease progression in PLHIV.

### *FCRL6* emerges as a promising biomarker of CMV seropositivity, evidenced by demethylation and increased expression and protein levels

Across multiple omics layers, we identified consistent markers that were significantly associated with CMV status in PLHIV in the discovery cohort, which were further replicated in the validation cohort (Supplementary Fig. S6A). Specifically, 508 markers were significantly validated at both methylation and gene expression levels, while 8 markers were even validated at plasma protein abundance levels (KIR2DL3, ITGAL, FCRL6, ADGRG1, GZMH, GBP1, GNLY, KLRD1) (Fig. 3, Supplementary Fig. S6B). KIR2DL3 is a natural killer (NK) cell receptor interacted with HLA-C1, and is strongly associated with the primary control of CMV infection in solid organ transplant recipients[43]. ADGRFG1 is a differentiation marker and inhibitory receptor on NK cells, and is expressed in a long-lived memory-like NK cell subset associated with prior human CMV infection[44].

Fc receptor-like 6 molecule (FCRL6) emerged as a significant signature associated with CMV serostatus at methylation (Supplementary Data S13), gene expression, and plasma protein abundances. Particularly, FCRL6 demethylation in CMV-seropositive PLHIV corresponds to upregulation in gene expression and plasma protein abundance (Fig. 3F, G). The association of FCRL6 with CMV seropositivity was also replicated in all the downsampling analyses (FDR < 0.05) (Supplementary Figs. S3–6). FCRL6 is an MHC class II receptor, and its expression in the peripheral blood of healthy individuals is mostly limited to T and NK cells[20]. Interestingly, whereas the MHC class II molecule HLA-DR is a ligand for FCRL6, CMV + PLHIV have an elevated number of HLA-DR+ immune cell subsets (Fig. 1E). The significant upregulation of FCRL6 expression on T cells of PLHIV, also in those with undetectable HIV viral load or normal CD4 cell counts, has been previously reported[20], including in hematopoietic stem cell-transplanted children with CMV reactivation[45]. Other markers associated with CMV serostatus at gene expression and protein levels, such as GNLY[46] are also related to cytotoxic T lymphocyte differentiation and functions (Supplementary Fig. S3). These findings show that FCRL6, together with GNLY, is upregulated at various biological layers in CMV-seropositive PLHIV, underscoring its potential as a comprehensive biomarker for the effect of CMV on the immune system.

To pinpoint which immune cells contribute to the elevated plasma FCRL6 signal, we performed multicolor flow cytometry on PBMCs from four PLHIV in our cohort (two CMV+ and two CMV-). Cells were stained for CD3, CD4, CD8, CD14, CD19, CD45, CD56, TCRγδ and FCRL6. CMV+ individuals showed higher proportions of FCRL6-expressing CD8 + T cells and γδ T cells than CMV- counterparts (Supplementary Fig. S8). The frequency of FCRL6+ cells was also increased within CD14+ monocytes and CD56+ natural-killer cells. These data confirm that multiple cytotoxic and innate subsets upregulate FCRL6 in vivo during CMV co-infection and provide a cellular source for the elevated protein we detect in plasma.

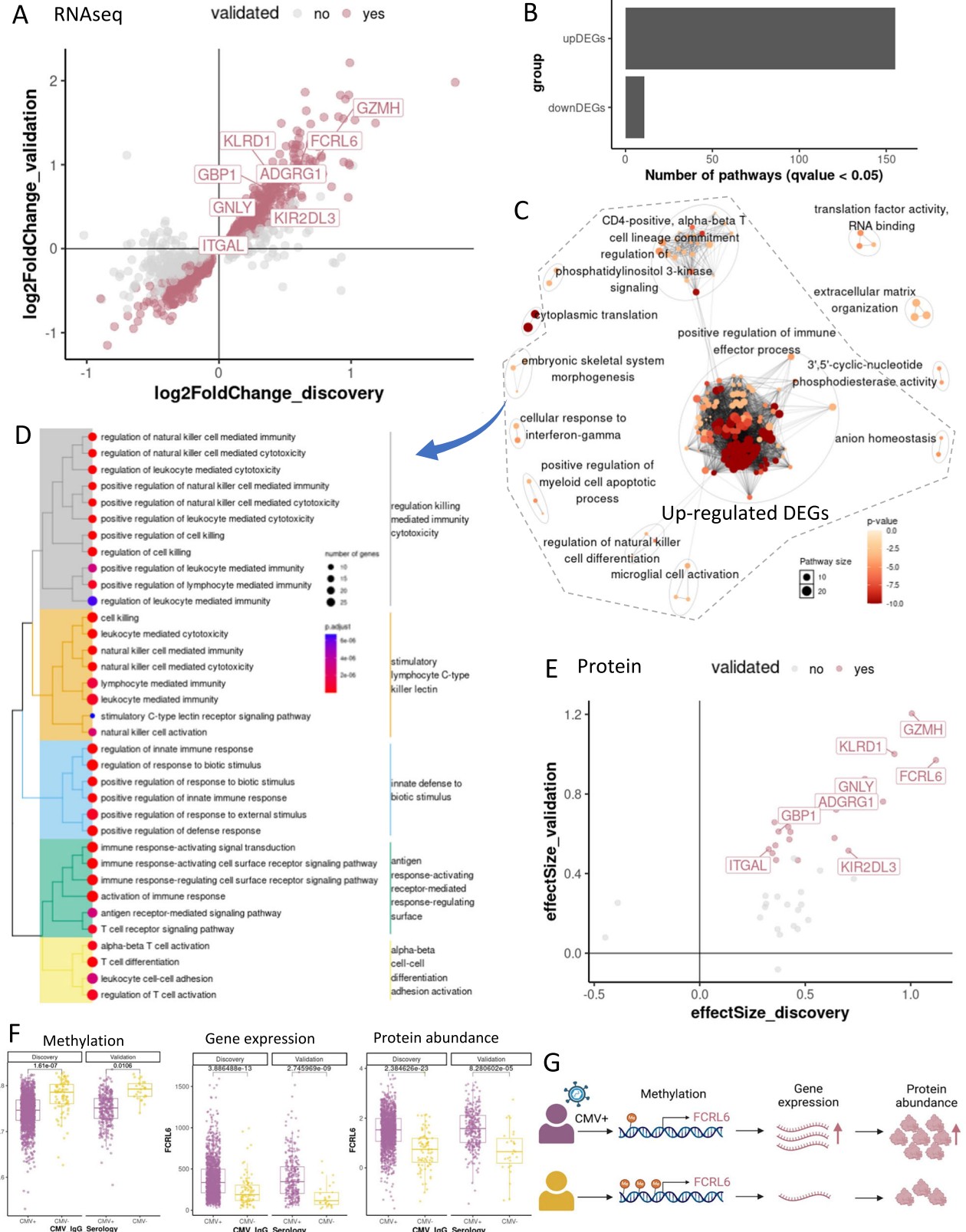

## GWAS identifies a significant CMV-seroprevalence locus linked to cytokine production and protein abundance

We hypothesized, therefore, the presence of a protective genotype and explored host genetics by performing a genome-wide association (GWAS) studies using 1338 donors of European ancestry in the discovery cohort (Supplementary Fig. S1A, B). The GWAS identified one

genome-wide significant locus on chromosome 15, with the leading variant rs7180928 (Fig. 4A; Supplementary Fig. S9A, B). Notably, the rs7180928-G allele confers protection against CMV seroprevalence in PLHIV, an effect also observed in the validation cohort (Fig. 4B).

Zooming into this locus to assess its regulation potentials, we evaluated *cis*-expression quantitative traits loci (*cis*-eQTL) and found

**Fig. 3 | CMV-associated signatures at the gene expression and plasma protein abundance. A** Gene expression signatures (DEG) which were significantly associated with CMV seropositivity. The dots present DEGs in discovery cohorts (FDR < 0.05). The purple color presents DEGs that were validated in the validation cohort (P < 0.05). We labeled some DEGs which were also cis-eQTM significant genes. **B** The number of sig. pathways for GO analysis with validated DEGs. **C** Pathway network of validated DEGs. **D** Pathway cluster of validated DEGs, which were upregulated in CMV-seropositive PLHIV. **E** Plasma protein signatures (DEP), which were significantly associated with CMV seropositivity. The dots present DEPs in discovery cohorts (FDR < 0.05). The purple color presents DEPs that were

validated in the validation cohort (P < 0.05). We labeled 8 DEPs, which were also validated DEGs. **F** The level of FCRL6 across omics layers (methylation, gene expression, protein levels; n = 1512 for discovery and 317 for validation cohort). In boxplots, center lines represent medians, boxes indicate 25th and 75th percentiles, and whiskers expand to 1.5× the interquartile range. Statistical tests were conducted using linear models adjusted for confounding effects, with two-sided testing and multiple-test adjustments by FDR. **G** Suggested scheme of FCRL6 regulation in CMV-seropositive PLHIV. Created in BioRender. Jiang, X. (2026) https://BioRender.com/6fxflef.

that the rs7180928-G allele was significantly associated with expression for neuronal acetylcholine receptor subunit beta-4 (CHRNB4) (P = 2.13e-05 by linear regression, Fig. 4C). However, the top cis-eQTL SNP of *CHRNB4* is rs11629784, but this SNP is excluded in the GWAS analysis due to low frequency (MAF = $6.96 \times 10^{-3}$) in the discovery cohort. To further confirm the role of *CHRNB4* in CMV infection in PLHIV, we performed colocalization and MR analyses to verify if the gene mediates PLHIV's resistance to CMV. The colocalization resulted in a PPH4 = 0.34, PPH3 = 0.23, PPH2 = 0.39, PPH1 = 0.02, and PPH0 = 0.03 (Bayesian factor analysis), suggesting a gentle but potential shared locus between *CHRNB4* expression and CMV- outcome. Subsequently, we performed a one-sample MR analysis to further support the observation using eQTL and GWAS summary statistics. The result indeed supports the colocalization analysis (Supplementary Fig. S9C, Supplementary Data S16). However, due to the limited expression of *CHRNB4* in whole blood, future work is required to confirm these observations (Supplementary Fig. S9D).

Furthermore, we estimated the correlation of the protective allele with immune-related molecules, including proteins and cytokines (Fig. 4D, E). Of note, the protective allele was associated with the decreased abundance of KIR2DS4 in the plasma (Fig. 4F, P = $1.3 \times 10^{-4}$, rho = −0.11, Spearman correlation test, one-way test, less than 0) and also correlated with the decreased IL22 production upon *M. tuberculosis* stimulation (Fig. 4G, P = $2.7 \times 10^{-2}$, rho = −0.07, Spearman correlation test, one-way test, less than 0). As KIR2DS4 was significantly highly expressed in CMV + PLHIV in the discovery cohort (P = $5.76 \times 10^{-3}$), we observed the same direction in the validation cohort (Supplementary Fig. S9E). Meanwhile, IL22 was also significantly higher in CMV+ compared to the CMV- PLHIV (P = $7.12 \times 10^{-5}$, Fig. 4G) and was replicated in the validation cohort (Supplementary Fig. S9F). To summarize, we identified a genome-wide significant locus associated with CMV seropositivity in PLHIVs; multiple QTLs integration analysis from the same cohort suggests that this locus is involved in the regulation of immune responses, potentially contributing to the susceptibility to CMV infection in PLHIV (Fig. 4J).

## Elevated gene and protein expression of FCRL6, driven by host genetics, is linked to CMV seropositivity

The consistency in FCRL6 expression associated with CMV seropositivity across multi-omics layers (Fig. 3F, G) prompted a hypothesis on whether gene expression and the plasma protein concentration of FCRL6 have a causal effect on CMV latent infection. We first performed QTL mapping for FCRL6 gene expression and protein abundance using linear regression models (Methods). Mendelian randomization (MR) leverages genetic variants (SNPs) associated with FCRL6 gene expression (eQTL) or protein abundance (protein QTL; pQTL) as instrumental variables (IVs) to infer causal relationships between these exposures and CMV seropositivity. We found significant positive causal relationships between both the gene expression (OR = 1.11; P = $2.98 \times 10^{-3}$) and protein abundance (OR = 0.66; P = $2.77 \times 10^{-4}$) with CMV infection risk (Fig. 5A). We primarily used the Inverse Variance Weighted (IVW) method to assess causal effects and conducted MR-Egger analysis to evaluate the robustness of these estimates against pleiotropy. The MR-Egger results for the protein

closely aligned with those from IVW, suggesting minimal pleiotropic bias (Fig. 5B, D). Given the broader (pleiotropic) effects of trans-QTLs, we filtered out genetic SNPs (instrumental variables) associated with multiple genes and proteins to address potential violations of MR assumptions (Fig. 5C, E). Notably, we observed no overlap between the IVs for FCRL6 gene expression in PBMCs (Fig. 5C) and those for protein abundance in whole blood (Fig. 5E). This is consistent with previous studies showing that eQTLs and pQTLs commonly involve distinct genetic variants, reflecting differences in post-transcriptional and translational regulation[47]. However, we did identify two SNPs located in close proximity on chromosome 14 (rs140233210 at position 20,061,131 for gene expression and rs1959651 at position 20,080,570 for protein expression). Although these SNPs are not in linkage disequilibrium (LD), their close genomic proximity suggests that some shared regulatory mechanisms influencing FCRL6 expression at different biological levels might exist, warranting further investigation.

FCRL6 is mostly present on mature NK cells and effector CD8 + T cells, and effector memory CD8 + T cells in healthy subjects, as well as on γδ T cells and some CD4 + T cells[48]. FCRL6 is known to interact with MHC-II/HLA-DR molecules, inducing tolerance, while its effect on cytotoxicity and cytokine production is less clear[49]. Non-classical HLA molecules can modulate NK-cell checkpoints during CMV infection, especially HLA-E, which presents CMV-derived peptides to NK-cell receptor NKG2C[50]. We therefore surveyed all omics layers for HLA-E signals. HLA-E mRNA in PBMCs was modestly higher in CMV+ individuals (P = 0.03, Supplementary Data S6), whereas circulating HLA-E protein and DNA methylation at the locus were unaltered. In the CMV GWAS, a sub-genome-wide signal lay adjacent to HLA-E (lead SNP rs2021368, p = $4.25 \times 10^{-5}$; Supplementary Fig. S9G). By contrast, whole-blood methylation profiling identified a robust CMV-associated CpG at the HLA-F/HLA-F-AS1 locus (Supplementary Data S14; cg08755130, FDR = $5.7 \times 10^{-7}$), hinting that epigenetic regulation of another non-classical HLA gene that engages NK receptors contributes to CMV-driven immune modulation[51,52]. Finally, we imputed classical HLA alleles with an HLA-specific reference panel to test whether MHC-I variation influences FCRL6 expression. We identified 4 HLA alleles that might affect the FCRL6 expression (P < 0.05); however, these alleles had not passed the multiple testing correction (Supplementary Data S15).

Together, these genetic, epigenetic, and expression data converge on a single theme: host-driven up-regulation of FCRL6, modulated further by variation in non-classical HLA genes, forms a key axis that governs susceptibility to latent CMV infection in antiretroviral-treated PLHIV.

## Discussion

This study provides a comprehensive multilayer evaluation of CMV-induced immune changes in 1887 antiretroviral-treated PLHIV. CMV seropositivity was linked to widespread DNA-methylation shifts and extensive gene expression changes, with protein-level alterations also evident but to a lesser degree. The most reproducible signal was FCRL6: its demethylated CpGs, higher mRNA abundance, and elevated plasma-protein levels were observed in both discovery and validation cohorts. *FCRL6* is preferentially expressed on mature NK cells and

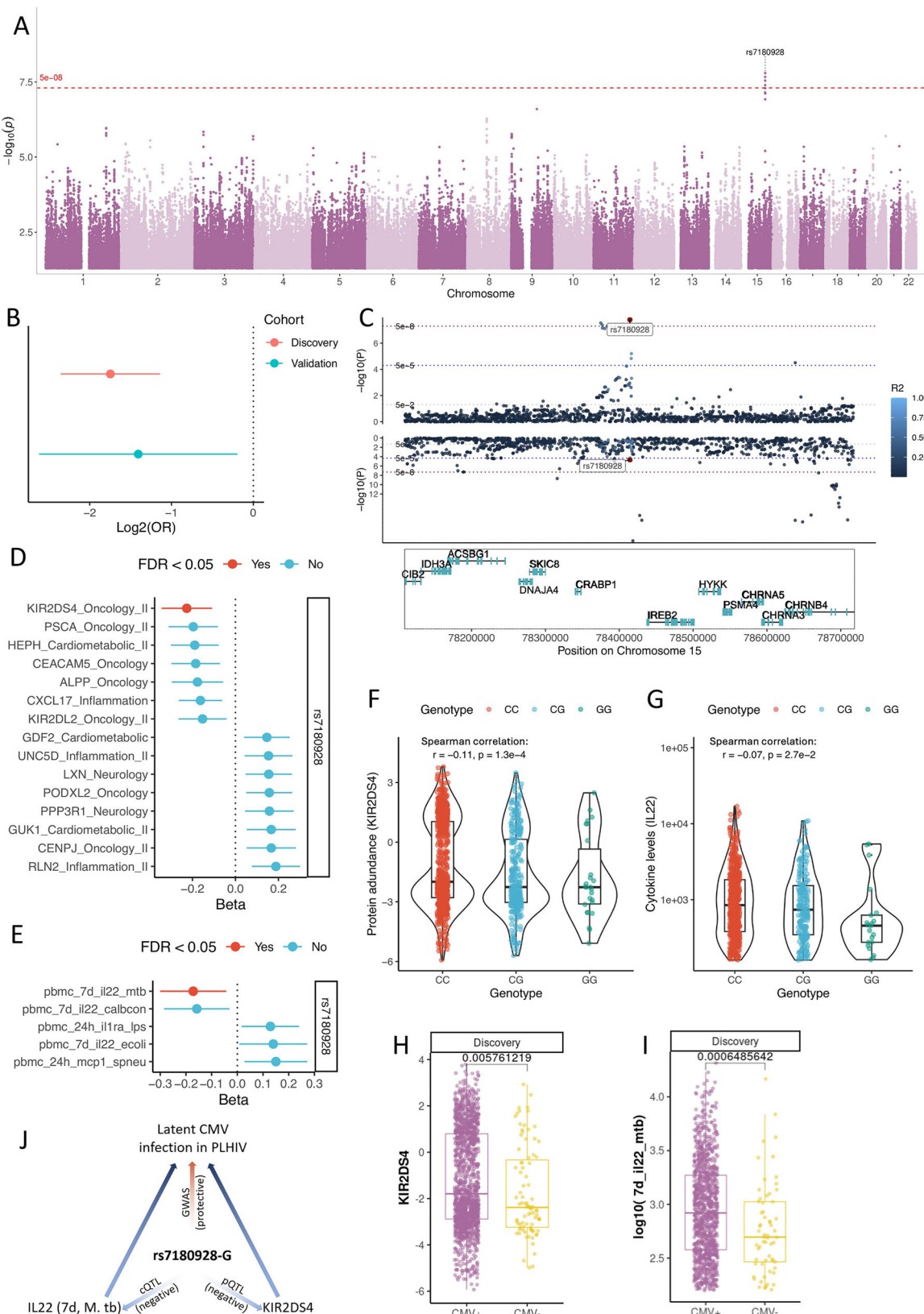

effector CD8 + T cells and binds HLA-DR, which we also found increased on CD4+, CD8 +, gamma-delta T, and NK cells in CMV-seropositive donors. MR analysis further suggests that genetically driven upregulation of *FCRL6* modestly raises the odds of CMV seropositivity, implying that FCRL6 is not only a market of past exposure, but may influence susceptibility to reactivation or disease. Lastly, the

genome-wide association analysis revealed a CMV-seroprevalence locus involving cytokine production and protein abundance.

Circulating immune cells of CMV + PLHIV produced more cytokines upon 24 h CMV stimulation, which may be due to either heterologous lymphocyte reactivity or CMV-induced alterations in innate immune cells. Prior research has indeed demonstrated the increased

**Fig. 4 | Genome-wide association study on CMV seropositivity in PLHIV.**
**A** Manhattan plot of CMV + /- in Europeans ($n = 1546$). *Y*-axis presents -log10-transformed *p*-values, and *X*-axis indicates SNP positions (GRCh38). The dashed red line is canonical genome-wide significant threshold (5e-8), and top SNP rs7180928 (chromosome 15) was labeled. **B** GWAS effects in discovery (red dot, $n = 1546$) and validation (blue dot) cohorts ($n = 322$). **C** Locuszoom plot shows genes in GWAS loci. Top and middle panels are genetic variants associated with CMV+/- outcomes or *CHRNB4* expression. Dots are SNPs and the blue color indicates LD R^2 (EUR) to rs7180928 (red). The bottom panel displays regional genes. **D** Protein QTL effect of rs7180928. The x-axis shows estimated regression coefficient of rs7180928-G ($n = 1064$). Color indicates FDR-corrected *p*-value < 0.05. **E** Cytokine QTL effect (rs7180928-G). The x-axis is the regression coefficient of rs7180928-G ($n = 1014$). Color indicates SNP passed threshold (FDR-corrected *p*-value < 0.05). **F** Violin plot showing correlations between genotypes and KIR2DS4 abundance ($n = 1342$). *Y*-axis indicates normalized protein abundance and *x*-axis shows genotypes (color as well). **G** Violin plot showing correlations between genotypes and IL22 level (7 days stimulated by *M. tuberculosis*, $n = 1342$). Y-axis indicates normalized cytokine levels (log10-transformed). *X*-axis and color show genotypes. **H** Boxplot shows abundances of KIR2DS4 between CMV + /CMV-PLHIV ($n = 1523$). **I** Boxplot shows levels of IL22 (7 days, M. tb) between CMV+ and CMV- PLHIV ($n = 1553$). **J** Schematic plot shows role of rs7180928 in latent CMV infection in PLHIV. Moreover, for boxplots (4f-i), whiskers expand to minima and maxima, boxes indicate 25th and 75th percentiles, centers represent medians. Data are presented as mean values +/- SD in Fig. 4b, d, e by adjusting multiple tests by FDR. Statistical tests are done by logistic regression for 4a and 4c, Spearman's correlation for 4f and 4g, t-tests for 4h and 4i. All statistical tests are two-sided with multiple-test adjustments by FDR.

pro-inflammatory cytokine production capacity of monocytes upon CMV exposure[53,54] as well as the reprogramming of monocytes by CMV, favoring a more proinflammatory phenotype[55]. We also observed a difference in phenotype of innate-like lymphocyte populations, such as a higher proportion of γδ T cells in CMV-seropositive PLHIV, which is in line with prior studies showing differences between CMV+ and CMV- bone marrow grafts[56]. Strikingly, we noticed the increased expression of HLA-DR in various immune cells, including CD4+ and CD8 + T cells, NK cells, and γδ T cells. HLA-DR is known to interact with FCRL6, a molecule that is expressed especially by mature T and NK cell subpopulations with cytotoxic potential.

Interestingly, *FCRL6* gene was significantly demethylated in CMV-seropositive PLHIV, which was in line with a significant elevation of *FCRL6* at transcriptome and proteome levels. As an MHC class II receptor, FCRL6 is mostly present on mature NK cells and effector CD8 + T cells and effector memory CD8 + T cells in healthy subjects, but also on γδ T cells and some CD4 + T cells[48]. FCRL6 is known to interact with MHC-II/HLA-DR molecules, inducing tolerance, while its effect on cytotoxicity and cytokine production is less clear[49]. Therefore, CMV-infected MHC-I expressing fibroblast, endothelial, or epithelial cells may be effectively eliminated by NK cells and CD8 + T cells, while CMV-infected MHC-II expressing myeloid cells are not targeted by NK cells, γδ T cells, and CD8 + T cells when expressing *FCRL6*. While the expansion of FCRL6-expressing cells, including CD4 + T cells, has been reported in PLHIV[57], our study indicated a dynamic relationship between *FCRL6* and CMV seropositivity, in which *FCRL6* is upregulated on immune cells in CMV + PLHIV, while high *FCRL6* could lead to higher susceptibility to CMV reactivation and CMV disease in PLHIV[57]. Our MR analysis even suggests elevated *FCRL6* expression as a causal factor contributing to CMV seropositivity. Downregulated FCRL6 expression may therefore prevent immune evasion and maintain cytotoxic properties so that CMV-infected cells can be eliminated.

Furthermore, our study demonstrates significant alterations in DNA methylation associated with CMV seropositivity, which is related to key immune pathways and cytokine responses. This highlights the CMV-induced epigenetic regulations in modulating immune function (Fig. 2H–J). Consequently, CMV seropositivity possibly upregulates genes involved in immune-mediated cytotoxicity, innate defense response, cytotoxicity, signaling, and regulation pathways at the transcriptome level in PLHIV. Hence, after the infection phase, CMV might continuously imprint its influence on both innate and adaptive immune cells in PLHIV. Among immune processes, we identified markers consistently associated with CMV seropositivity across multi-omics layers, including KIR2DL3[43] and ADGRFG1[44] related to NK cell functions and differentiation, and GNLY[46] related to cytotoxic T lymphocyte functions and differentiation. Cytotoxic T lymphocytes play a crucial role in host defense against CMV; together with FCRL6, the upregulation GNLY in CMV-seropositive PLHIV at various biological layers suggests their potential as biomarkers of the effect of CMV on the immune system.

Our GWAS identified a genome-wide significant locus on chromosome 15, with the rs7180928-G allele linked to protection against CMV seroprevalence. Among genes near this locus, a more than two-fold increase of *CHRNB4* expression was noticed after treatment by anti-CMV peptide antibodies[58], indicating the potential role of this gene in CMV infection. Furthermore, this gene plays a role during infection of the Zika virus[59], and was also linked to smoking and nicotine dependence[60], suggesting its pleiotropic potential. The rs7180928-G allele identified was also negatively associated with IL22 production (7 days, *M. tuberculosis* stimulation) and KIR2DS4 plasma abundance. Different studies in transplant recipients indeed showed a positive association between KIR2DS4 and CMV reactivation[61]. The KIR2DS4 protein induces the activation of NK cells upon binding by specific peptides from HLA-C, while the long cytoplasmic tail isoform of KIR2DS4 does not activate the NK cells[62]. These findings suggest that this locus is involved in immune response regulation and speculation to CMV in PLHIV, which offers insights into genetic factors influencing viral infections in this population and suggests avenues for further research in immune response modulation. Since our cohort is mainly European participants, we still need to examine this locus in other ethnic groups.

Previous studies have suggested that genetic factors are potentially involved in the CMV infection in different disease contexts, such as individuals after allogeneic hematopoietic cell transplantation and individuals who are diagnosed Schizophrenia[63–65]. However, the potential roles of genetic variants in resistance to CMV infection in PLHIV are less known, in contrast to the observed high prevalence of CMV seropositivity. Therefore, we performed a GWAS to screen genetic loci that are associated with CMV in our PLHIV cohort. The screening suggests a locus on chromosome 15 where the rs7180928-G allele is protective, which is not previously reported. To confirm its role in CMV infection in our study cohort, we systematically estimated whether this locus bridges other molecular traits with CMV-sero-negativities in PLHIV. The analysis suggested interplays among the identified locus, cytokine production, protein abundance, and gene expression, which collectively highlighted the importance of the identified locus in CMV resistance upon HIV-positive.

CMV seroprevalence in the 2000HIV study is around 94%, which is considerably higher than the approximately 50% reported for age-matched adults in the general Dutch population[66]. This elevation almost certainly reflects the cohort's demographic composition, which is predominantly middle-aged men who have sex with men (MSM) living with HIV and using long-term ART. High baseline prevalence reduces statistical power to study CMV- PLHIV and may exaggerate the magnitude of CMV-associated phenotypes. However, the down-sampling analyses showed high concordance with the full-cohort results, with the vast majority of significant associations replicated across iterations, demonstrating robust reproducibility. Even so, the cytokine and multi-omic signatures reported here should be interpreted with caution in settings where CMV exposure is less common,

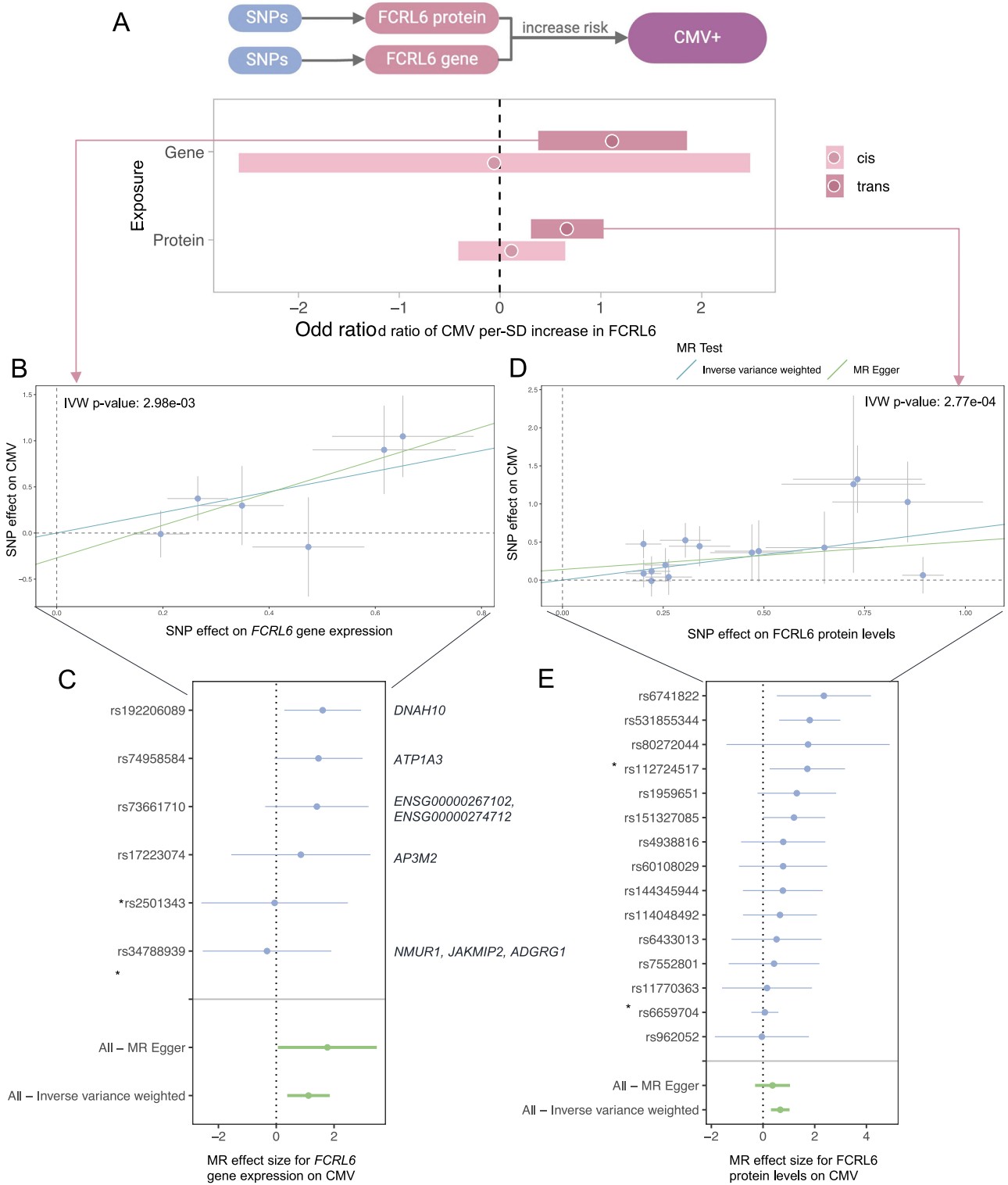

**Fig. 5 | Mendelian Randomization (MR) analyses evaluating the causal relationship between FCRL6 and CMV seropositivity in people living with HIV.**
**A** Inverse Variance Weighted (IVW) causal effect estimates (log-odds ratio of CMV per 1-SD increase in FCRL6) ± SE for genetically predicted *FCRL6* gene expression and circulating FCRL6 protein, using cis and cis+trans instruments. All MR analyses were performed using summary-level genetic association statistics derived from independent biological samples (eQTL $n = 1048$, pQTL $n = 1064$, CMV serostatus GWAS $n = 1076$). **B** Scatter plot of SNP instruments for the FCRL6 gene-expression trans-MR, showing SNP effects on expression (x-axis) ± SE versus CMV (y-axis) ± SE.

Including IVW (blue) and MR-Egger (green) regression lines (IVW $P = 2.98 \times 10^{-3}$). **C** Forest plot of single-SNP causal estimates for the gene-expression MR, with SNP-specific IVW estimates ± SE (blue) and combined IVW and MR-Egger effects (green). Gene labels show other genes the SNP is associated with ($P < 1 \times 10^{-5}$). Asterisks denote cis-SNPs. **D** Scatter plot of SNP instruments for the protein trans-MR, displaying SNP effects on FCRL6 protein (x-axis) ± SE and CMV (y-axis) ± SE, with IVW and MR-Egger regression lines (IVW $P = 2.77 \times 10^{-4}$). **E** Forest plot of single-SNP estimates for the protein MR, with SNP-specific IVW estimates ± SE (blue) and combined IVW and MR-Egger effects (green). Asterisks denote cis-SNPs.

and future multi-center studies that include low-prevalence populations will be required for external validation.

In conclusion, we identified robust molecular signatures related to CMV seropositivity and genetic signatures related to potential CMV susceptibility in PLHIV. This study also emphasized the importance of investigating multi-omics signatures to understand the impact of CMV seropositivity on human immunity. Studying the role of *FCRL6* in CMV infection will provide further insights into the impact of CMV co-infection in PLHIV and suggestive markers to modulate CMV-related complications in PLHIV. Additionally, the impact of CMV seropositivity in PLHIV can be projected to the CMV infection in other immuno-compromised individuals, such as transplant recipients.

## Methods

### Study design and data sources
The 2000HIV study is a prospective multicentric observational longitudinal cohort of PLHIV on stable cART[10]. Participants were recruited from October 2019 until October 2021. The 2000HIV study was approved by the Independent Review Board Nijmegen (NL68056.091.81) and published at ClinicalTrials.gov (NCT03994835). Written informed consent was received from participants before inclusion in the study. All experiments with human samples were conducted following the principles of the Declaration of Helsinki. The sample sizes were comprehensively discussed in our previous publication, which announced the 2000HIV cohort[10]. The cohort was made up of the discovery cohort and the validation cohort. Participants in the discovery cohort were recruited from three specialized Dutch HIV treatment centers, two university medical centers, and one large general hospital (Radboudumc Nijmegen, Erasmus MC Rotterdam, and OLVG Amsterdam). Participants in the validation cohort were recruited in a separate medical center, a large general hospital (Elisabeth-TweeSteden Ziekenhuis Tilburg). Although the samples of the two sub-cohorts were collected separately, processing and measurements were identical. Inclusion criteria were: proven HIV-1 infection, age of 18 years or older, receiving cART for at least 6 months, and with a latest HIV-1 RNA of less than 200 copies/mL. Additionally, individuals with spontaneous control of HIV-1 without cART could participate if viral loads had been lower than <10,000 copies/mL for at least 5 years, during which CD4 + T-cell counts were stable >500 cells/mm3. Exclusion criteria were: no informed consent, insufficient communication because of language barriers or other problems, current pregnancy, detectable viral hepatitis B or C DNA by polymerase chain reaction (PCR), or signs of any current acute infection. In the 2000HIV cohort, we assessed the CMV status of 1887 PLHIV by performing an ELISA assay on serum samples according to the manufacturer's protocol (GenWay, San Diego, California, US) (Supplementary Data S1).

### Ex-vivo cytokine production measurement and analysis
Peripheral blood mononuclear cells (PBMCs) were stimulated with a panel of whole (inactivated) pathogens, pattern recognition receptor ligands, pathogen-derived antigens, and viral stimuli to assess cytokine production capacity. Stimulations were performed in round-bottom 96-well plates (Greiner Bio-One) using 500,000 cells per well. Cells were incubated either for 24 h at 37 °C and 5% $CO_2$ or for 7 days in the presence of 10% human pooled serum.

For the 24-h stimulations, the stimuli included poly-inosinic:polycytidylic acid (poly I:C), lipopolysaccharide (LPS), imiquimod, interleukin-1α (IL-1α), HIV viral envelope, cytomegalovirus (CMV), and Streptococcus pneumoniae. The cytokines measured in supernatants after 24 h were IL-1β, IL-1Ra, IL-6, IL-8, IL-10, MCP-1, MIP-1α, and TNF. For the 7-day stimulations, the stimuli included Escherichia coli, Staphylococcus aureus, Streptococcus pneumoniae, Mycobacterium tuberculosis, Candida albicans (conidia and hyphae), and phytohemagglutinin (PHA). The cytokines measured after 7 days were IL-5, IL-10, IL-17, IL-22, and IFNγ. After stimulation, supernatants were collected and stored at −20 °C until cytokine concentrations were quantified using enzyme-linked immunosorbent assay (ELISA). Details regarding stimulant concentrations, manufacturers, and microbial strains are provided in the cohort description.

The cytokine production abundance underwent inverse rank transformations to normalize their distributions across samples. We identified cytokine with differential production abundance between CMV-seropositive (CMV+) and CMV-seronegative (CMV-) individuals in the discovery cohort (FDR < 0.05) using Robust Linear Regression model (RLM) with adjustment for the confounding effect of age, sex, BMI, center, season (season_sin, season_cos), and ethnicity (top5 genetic PCs) for the discovery cohort. We then validated the significant cytokine production abundances in the validation cohort. Since the validation cohort had a smaller sample size and was collected in one center site, we performed a similar differential cytokine production abundance analysis between CMV+ and CMV- individuals in the validation cohort ($P < 0.05$) with adjustment for the confounding effect of age and sex. Confounders were selected based on prior knowledge and by inspecting the leading principal components in Principal Component Analysis (PCA). The cytokine production upon stimulation, which had significant differences between CMV+ and CMV- and had the same direction in the effect size in both discovery and validation cohorts, was considered significantly replicated.

### Immune cell profiling and analysis
Whole blood samples were immunophenotyped by using three flow cytometry panels containing 17–20 markers each and custom-made tubes with dry antibodies from DURA Innovations Technology (Beckman Colter). Cells were acquired in a 21-color, six-laser CytoFLEX-LX (Beckman Colter) and using Cytexpert software 2.3 (Beckman Colter). Instrument quality control and standardization were performed daily using CytoFLEX Daily QC Fluorospheres (Beckman Colter catalog #B53230), CytoFLEX Daily IR QC Fluorospheres beads (Beckman Colter catalog # C06147), and SPHEROtm Rainbow calibration particles 6-peak (Spherotech Inc, catalog # RCP-30-5A-6) (21). Data analysis was performed using Kaluza V2.1.2 and Cytobank Platform V9.0 (Beckman Colter)[10].

The absolute cell count underwent inverse rank transformations to normalize their distributions across samples. We identified cell subsets with different cell counts between CMV+ and CMV- individuals in the discovery cohort (FDR < 0.05) using RLM models with adjustment for the confounding effect of age, sex, BMI, center, season (season_sin, season_cos), and ethnicity (top5 genetic PCs) for the discovery cohort. We then validated these differential cell count subsets in the validation cohort. Since the validation cohort had a smaller sample size and was collected in one center site, we performed a similar cell count analysis between CMV+ and CMV- individuals ($P < 0.05$) with adjustment for the confounding effect of age and BMI. Confounders were selected based on prior knowledge and by inspecting the leading principal components in PCA. The cell type subsets, which were significantly different in cell counts between CMV + and CMV- individuals and had the same direction in the effect size in both discovery and validation cohorts, were considered significantly replicated.

### Methylation measurement and analysis
DNA methylation was performed on a total of 1914 samples. DNA was isolated from EDTA whole blood by the Radboudumc Genetics Department using ChemagicStar automated configuration (consisting of the Microlab STAR and Chemagen Magnetic Separation Module 1, Hamilton Robotics), combined with Chemagen nucleic acid extraction technology with magnetic polyvinyl alcohol (M-PVA) beads, which follows a standard and automated bind-wash-elute procedure. The concentration of the DNA and 260/280 nm ratio were determined using a NanoDrop spectrophotometer, after which samples were

normalized to 50 ng/μL in TE buffer and randomly distributed amongst plates. High-quality DNA was selected for genome-wide DNA methylation profiling using the Illumina Infinium MethylationEPIC BeadChip array (MethylEPIC v1 manifest B5). Standard sample- and probe-based quality control were performed.

As previously described, the DNA methylation dataset was divided into a discovery cohort ($n = 1546$) and a validation cohort ($n = 322$), and each cohort was analyzed separately. DNA methylation values were estimated from the raw IDAT files using the minfi package in R (v.4.2.0). Preprocessing steps were done to discard two gender mismatch samples from the discovery cohort, one bad-quality sample from the validation cohort (call rate <99%). Probes (Discovery: $n = 2743$ and Validation: $n = 2641$) with methylation value missing (detection $P > 0.01$) at >10% samples and probes within the sex chromosome ($n = 19,627$) were also excluded from the downstream analysis. Since the majority of the participants are European, we also removed the probes containing SNPs at the target CpG sites with a MAF > 5% in European populations, as well as the probes that mapped to multiple loci, i.e., polymorphic probes, as suggested in (Both Discovery and Validation: $n = 52,173$). Next, we implemented stratified quantile normalization. Methylation value was also utilized for estimating the proportion of six immune cell types, namely neutrophils, monocytes, B-Cells, NK cells, CD8-T cells, and CD4-T cells, using modified Housman's method available within the estimateCellCounts2 function of the FlowSorted.Blood.EPIC package of R. Methylation $\beta$-values was calculated as a percentage: $\beta = M/(M + U + 100)$, where M and U represent methylated and unmethylated signal intensities, respectively, and $\beta$-values were then transformed to M-values as $\log2(\beta/(1 - \beta))$, and M-values were used in all downstream analyses.

To mitigate the effect of extreme outliers in data, we trimmed the methylation set using: (25th percentile − 3*IQR) and (75th percentile + 3*IQR), where IQR = interquartile range. Differentially methylated CpG sites associated with latent CMV infection by fitting a robust linear regression model. The methylation M value was used as the outcome variable, and the model was corrected for age, sex, season effect (using $\sin(2\pi t/T)$ and $\cos(2\pi t/T)$ (t is the time, T is the period), batch effects, and immune cell proportions. For EWAS with all ethnicity, the top five PCs extracted from the genotype of the same individuals were included in the model for the correction of ethnicity. For validation, only age and sex were adjusted in the model.

To assess the relationship between DNA methylation and gene expression, we calculated the Pearson correlation coefficient (r) between methylation levels at specific CpG sites and the expression of corresponding genes. The correlation analysis was performed using the cor() function in R, and statistical significance was evaluated using cor.test(). The correlation coefficient (r) and associated p-values were used to determine the strength and direction of the association.

CpGs were considered significantly replicated only if they have (i) discovery cohort: FDR < 0.05, (ii) validation cohort: same direction as the discovery cohort, and FDR < 0.05. eQTM was performed to explore the correlation between CpG methylation and gene expression. Replicated CpGs associated genes were subjected to enrichment analysis using the clusterProfiler package in R. Cis-eQTM was performed by the Limma[67] package. Genes within 250 k up- and downstream of the CpG sites were tested. The mediation effects of the CpG sites were tested by the mediation R package[29]. The threshold for partial mediation is *EffectSize* of *prop_mediated* greater than 0.2.

## Gene expression measurement and analysis
Bulk RNA-seq of PBMCs was performed on Illumina platforms (>30 million reads per sample). Sequencing reads were aligned to the human reference genome (NCBI build 38) using STAR. Key QC metrics (read quality, duplication rates, alignment rates, gene-body coverage, etc.) were aggregated via MultiQC, and all samples passed standard thresholds. We then applied the following sample-level filters to the initial set of 1899 libraries: (i) excluding 2 libraries per Nijmegen team decision, (ii) removed 1 library with no uniquely mapped reads, (iii) discarded 3 duplicate-donor libraries, retaining the replicate with higher read depth, (iv) excluded 29 libraries with <5 million uniquely mapped reads, (v) excluded 7 libraries with discordant reported sex versus Y-chromosome read counts. This resulted in 1857 final samples (Supplementary Fig. S10). Importantly, no samples were removed as PCA outliers, in order to retain biologically relevant variation (Supplementary Fig. S10). For gene-level QC, 58,347 transcripts were detected, of which those with >=10 counts in the smallest comparison group were retained, yielding 16,000 to 20,000 genes per analysis. The reads underwent rlog transformation and normalization using the DESeq2 package[68].

After normalization, we identified differentially expressed genes between CMV+ and CMV- individuals in the discovery cohort (FDR < 0.05) using the DESeq2 package[68] with adjustment for the confounding effect of age, sex, BMI, center, season (season_sin, season_cos), and ethnicity (top5 genetic PCs). We then validated these differentially expressed genes in the validation cohort. Since the validation cohort had a smaller sample size and was collected in one center site, we performed a similar differential gene expression analysis between CMV+ and CMV- individuals ($P < 0.05$) with adjustment for the confounding effect of age and season (season_sin, season_cos). Confounders were selected based on prior knowledge and by inspecting the leading principal components in PCA. The gene expression, which were significant differences between CMV+ and CMV- and had the same direction in the log fold change in both discovery and validation cohorts, was considered significantly replicated.

## Plasma protein measurement and analysis
Circulating plasma protein expression was assessed with the use of a commercially available multiplex proximity extension assay (PEA) from Olink® proteomics AB (Uppsala, Sweden). Plasma samples from PLHIV were measured in three batches. The first batch ($n = 692$ samples) was measured using the library Olink® Explore 1536, consisting of 1472 proteins divided into four 384-plex panels focused on inflammation, oncology, cardiometabolic, and neurology proteins (panels I). The second batch ($n = 692$ samples) was measured using the Olink® Explore Expansion 1536, consisting of 1472 proteins divided into four 384-plex panels focused on additional inflammation II, oncology II, cardiometabolic II, and neurology II proteins (panels II). The third batch was measured using the full library (Olink® Explore 3072), consisting of ~3000 proteins divided into eight 384-plex panels focused on inflammation, oncology, cardiometabolic, and neurology proteins (panels I and II). Protein measurements are delivered as Normalized Protein expression (NPX) values, which are Olink's relative protein quantification unit on a log2 scale. Olink has developed a built-in quality control (QC) system using internal controls to control the technical performance of assays and samples. In addition, Olink services performed bridging normalization between oncology panel II and neurology panel II from batches two and three using 27 bridging samples. In addition to Olink's QC, we performed bridging normalization to remove the batch effect between inflammation, oncology, cardiometabolic, and neurology panels from batch one and three using 32 bridging samples. In addition, we performed bridging normalization between cardiometabolic panel II and inflammation panel II from batches two and three using 23 bridging samples. Bridging normalization was performed by following the next steps for each protein: (1) We first calculated the median of the bridging samples for each protein in the two batches. (2) Then, we calculated the median difference, keeping one batch as a reference. (3) We finally subtracted the median difference from each protein in the non-reference batch. For bridging normalization between batch one and three, we used 1461 unique proteins measured in 692 samples that were common in batch one and three. For normalization between cardiometabolic II and

inflammation II panel from batch two and three, we used 736 proteins measured in 1268 samples. Limit of detection (LOD) values per protein were re-adjusted by the same adjustment factor as the respective protein measurements after bridging normalization.

After removing batch effect using bridging normalization, we performed standard quality control per protein and sample. In each of the eight panels from the Olink@Explore 3072 platform, IL6, TNF, CXCL8, LMOD1, SCRIB, and IDO1 were measured as technical duplicates for quality control purposes. Strong correlations were observed between the technical duplicates among panels, and therefore, we selected the measurements from the inflammatory panel. Next, we excluded proteins with LOD >= 25 of the samples ($n = 547$ proteins were excluded), resulting in 2367 proteins for follow-up analysis. During QC per sample, we excluded 20 samples that were recorded as taking immunomodulatory drugs. Finally, to detect outliers, we performed PCA using the NPX values from 1938 samples. Outliers were defined as those samples falling above or below four standard deviations (SD) from the mean of principal component one (PC1) and/or two (PC2). In total, seven samples were excluded based on PCA, resulting in 1910 samples.

The plasma protein abundance underwent inverse rank transformations to normalize their distributions across samples. We identified proteins with differential abundance profiles between CMV+ and CMV- individuals in the discovery cohort (FDR < 0.05) using RLM models with adjustment for the confounding effect of age, sex, BMI, center, season (season_sin, season_cos), and ethnicity (top5 genetic PCs). We then validated these differential abundance proteins in the validation cohort. Since the validation cohort had a smaller sample size and was collected in one center site, we performed a similar differential abundance analysis between CMV+ and CMV- individuals ($P < 0.05$) with adjustment for the confounding effect of sex and season (season_sin, season_cos). Confounders were selected based on prior knowledge and by inspecting the leading principal components in PCA. The protein abundances, which were significant differences between CMV+ and CMV- and had the same direction in the effect size in both discovery and validation cohorts, were considered significantly replicated.

For a comparative analysis in healthy individuals, we used Olink data from the Dutch 300BCG study[38] (CMV+ = 74, CMV- = 241). Out of the 273 significantly differentially expressed proteins in the PLHIV discovery set, nine were also measured in the inflammatory panel used for this cohort. For these nine proteins, we fitted a linear model, correcting for age and sex effects.

## Plasma metabolite measurement and analysis

We measured the abundance of 1720 circulated metabolites in 1902 serum samples by using General Metabolic's untargeted metabolic platform. Untargeted metabolome profiling was performed on plasma samples using flow injection electrospray - time-of-flight mass spectrometry in collaboration with General Metabolics LLC[69]. Metabolites were identified based on the mass-to-charge ratio (ion m/z). The duplicate peak intensities in raw metabolome data were averaged and normalized using moving median normalization before further analysis. Principal component analysis was then applied to identify possible outlier samples. Metabolites are annotated and further categorized based on metabolomic source (endogenous, food, or drugs) and chemical taxonomy using publicly available data from The Human Metabolome Database[70].

According to the Human Metabolome Database (HMDB), 851 out of these 1720 metabolites are endogenous metabolites and were used for further analysis. The plasma metabolite abundance underwent inverse rank transformations to normalize their distributions across samples. Thereafter, we identified metabolites with differential abundance profiles between CMV+ and CMV− individuals in the discovery cohort (FDR < 0.05) using RLM models with adjustment for the

confounding effect of age, sex, BMI, center, season (season_sin, season_cos), and ethnicity (top5 genetic PCs). We then validated these differential abundance metabolites in the validation cohort. Since the validation cohort had a smaller sample size and was collected in one center site, we performed a similar differential abundance analysis between CMV+ and CMV- individuals ($P < 0.05$) with adjustment for the confounding effect of age and season (season_sin, season_cos). Confounders were selected based on prior knowledge and by inspecting the leading principal components in PCA. The metabolite abundances, which were significant differences between CMV+ and CMV- and had the same direction in the effect size in both discovery and validation cohorts, were considered significantly replicated.

## Downsampling analysis

Since there is a limited number of CMV individuals ($n = 86$) in the discovery cohort, to assess whether our larger CMV+ sample introduced bias, we repeated the seropositivity association analysis 100 times on downsampled datasets for DNA methylation, gene expression, and protein abundance. In each iteration, we randomly selected 86 CMV+ individuals to match the existing 86 CMV- individuals in the discovery cohort ($n = 172$). We then compared the effect sizes from these 100 downsampled analyses to those from the original full-cohort analysis ($n = 1553$). Finally, we calculated the proportion of molecules (FDR < 0.05) whose associations were consistently replicated across the downsampling iterations.

## Cross-referencing CMV-responsive signatures with published MOFA latent factors

To assess the role of CMV-associated molecules, we utilized a previously published Multi-Omics Factor Analysis (MOFA) model of the European discovery subset ($n = 996$), which integrated matched whole-blood RNA-seq, plasma proteomics (Olink), untargeted metabolomics, and ex vivo cytokine responses[39]. The complete factor-feature weight matrix from that study served as the reference for all enrichment analyses described below; no additional factor training or tuning was performed in the present work.

Within four latent factors linked to clinical phenotypes, loadings were ranked separately by absolute positive and negative weights. Based on the direction of the correlation to the clinical phenotype, we labeled loadings as *risk* or *protective* for the respective disease. E.g., if the LF correlated positively with the phenotype, positive weights were deemed *risk*, if the LF correlated negatively, negative weights were deemed *risk*.

Eleven percentile cut-offs (5%-95% in 10% steps) defined nested top-N feature lists for both directions. At each cut-off, we performed one-tailed hypergeometric tests comparing the selected features against the corresponding CMV discovery lists, restricted to the view-specific universe. Four tests were evaluated per percentile (risk-up, risk-down, protective-up, protective-down). Benjamini-Hochberg correction was applied per LF across all percentiles and test combinations. Significant enrichments (adjusted $P < 0.05$) were depicted as dot plots: point size represents percent overlap, color encodes -log10(padj), facets separate risk vs protective weights and CMV+/CMV- discovery sets.

## Genotyping, imputation, and quality control

DNA was extracted from the whole blood of each participant. Genotyping for the multi-ethnic 2000HIV cohort was performed using the Illumina Infinium Global Screening Array. To ensure quality, raw variants and samples underwent rigorous quality control using PLINK v1.90b[71]. Variants with over 5% missing genotype data and those deviating from Hardy-Weinberg equilibrium (HWE) with a $P$ value $< 1 \times 10^{-6}$ were excluded. The HWE exact test was conducted on variants categorized by ethnicity. Samples with a call rate below 97.5% and those with heterozygosity rates deviating more than three

standard deviations (SD) from the mean for their reported ethnicity were removed. Variants that passed quality control were mapped from GRCh37 to GRCh38 using the UCSC liftOver tool. Subsequently, TOPMed Freeze5 on genome build GRCh38 was utilized to align strands to the TOPMed reference panel. The McCarthy group's tools were employed for alignment. Post quality control, 582,404 variants from 1864 individuals were retained for imputation. These filtered variants were uploaded to the TOPmed Imputation server and matched against the TOPMed (version r2 on GRCh38) reference panel. Using BCF tools, the imputed variants were filtered by ethnicity, excluding those with low imputation quality scores (R2 < 0.3 or ER2 < 0.7) and MAF < 1%, resulting in 10,810,841 variants from 1864 participants in the 2000HIV multi-ancestry cohort.

### Genome-wide association study of CMV infection

To understand the genetic determinants of CMV infection in the current study cohort, we performed a GWAS analysis using genome-wide SNPs (*Genotype and quality control*) to explain the CMV serostatus, i.e., negative or positive. The GWAS analysis follows the standard discovery-validation strategy, where the discovery and validation cohorts include 1546 and 322 donors, respectively. The GWAS was performed using the PLINK tool (version 1.90b) with the following options: *--glm allow-no-covars firth-fallback --1 --covar-variance-standardize --no-sex --maf 0.05*. Age, gender, ethnicity, BMI, smoking, risk behaviors, CD4 nadir, CD4/CD8 ratio, HIV duration, cART duration, center, and top 20 genotype PCs (calculated by PLINK using default parameters) were included as covariables to account for potential confounding effects. The obtained summary statistics were visualized by a Manhattan plot to show the identified loci at $p$-value threshold of $5 \times 10^{-8}$ using the R package topr (version 1.1.0). A fair lambda value (1.08) of the summary statistics indicate less risk of inflation, which was visualized by QQ plot using tpor (Supplementary Fig. S9B). The top association signal was explored by regional plot using topr with default parameters to prioritize potential causal variants and genes. The obtained summary statistics are publicly available on Zenodo (https://doi.org/10.5281/zenodo.12580771).

### Quantitative Loci (QTL) mapping

Following imputation, PLINK 2.0[72] was used for a final round of quality control. Variants failing the HWE test at $P < 1 \times 10^{-12}$, those with a MAF < 1%, and an R2 < 0.05 were excluded. After this process, 8,944,122 imputed SNP variants were retained for subsequent analysis.

We performed QTL mapping for gene expression ($n = 1048$), protein levels ($n = 1064$), and cytokine response ($n = 1014$) using the MatrixEQTL R package (v2.3)[73]. We applied a linear model to map the inverse-rank transformed values to the genotype data, accounting for age, sex, BMI, seasonality, pre-COVID-19 inclusion, COVID-19 vaccination status, and recruitment center. Seasonality was modeled using a sine and cosine wave with a period of 365.25 days, which can combine to form a sine wave with any phase and a yearly frequency[74]. The full summary statistics can be found at https://lab-li.ciim-hannover.de/apps/hiv_xqtl_atlas/.

### Mendelian randomization

We performed Mendelian Randomization (MR) analyses to explore the association between FCRL6 gene expression and protein levels (exposures) with latent CMV infection (outcome) using TwoSampleMR (v0.5.7)[75]. For cis-MR, instrumental variables (IVs) were selected within a 2 Mb window around the *FCRL6* gene with stringent clumping (kb = 10,000, $r^2 = 0.001$) and a significance threshold of $P < 1 \times 10^{-3}$, yielding one significant SNP for gene expression and two for protein levels. Additionally, we identified 19 and 17 trans-IVs ($P < 1 \times 10^{-5}$) for gene expression (eQTL) and protein levels (pQTL), respectively. To ensure specificity and reduce pleiotropy, we retained only trans-eQTLs associated with fewer than five genes ($P < 1 \times 10^{-5}$). For trans-pQTLs, we

retained those exclusively associated with the FCRL6 protein ($P < 1 \times 10^{-5}$), yielding 14 SNPs. The cis SNPs were also included in the trans-MR analysis. We applied the Wald-ratio for a single instrumental SNP and inverse-variance weighted (IVW) for multiple SNPs, followed by sensitivity checks for heterogeneity ($P > 0.05$) and pleiotropy ($P > 0.05$), and a leave-one-out analysis (maximum $P < 0.05$) to assess the robustness of the findings. This comprehensive approach allowed for a careful assessment of the genetic drivers of CMV in the context of *FCRL6* expression and its protein levels.

### MHC genetics related to FCRL6 expression analysis

The Michigan imputation server was used to impute SNPS, alleles, and amino acids in the MHC region (6:27,970,031-33,965,553) on GRCh37 build using the four-digit multi-ethnic HLA v2 reference panel (ref). After the standard QC per SNP and sample as described above, genotyped variants from chromosome 6 from 1864 individuals of all ethnicities from the 2000HIV cohort were submitted for imputation. Genetic variants were lifted from GRCh38 to GRCh37 using the Michigan server. After imputation, we obtained 22,733 SNPs, alleles, and amino acids within the MHC region, of which 570 were HLA classical alleles at two and four-digit resolution (HLA-A, HLA-B, HLA-C, HLA-DQA1, HLA-DQB1, HLA-DPA1, HLA-DPB1, HLA-DRB1) and 3449 amino acids, 4023 SNPs within HLA, and 14691 scaffold SNPs.

Then, we mapped expression-QTLs (eQTLs) using MHC markers (HLA alleles, imputed amino acids, SNPs within the MHC region, etc, in total 19792 markers) with a minor allele count of at least 10 (MAC > 10) in the discovery cohort and the gene counts of the FCRL6 gene. HLA alleles at two-digit resolution were removed from eQTL mapping. In total, we used 986 samples of European ancestry from the discovery cohort and 251 samples from the validation cohort that have both imputed genetic and transcriptomic data.

We applied a linear model to map the FCRL6 gene counts to the MHC data, accounting for age, sex, BMI, seasonality, and the first five PCs to account for population stratification. The gene counts were transformed using variance stabilizing transformation (VST) as implemented in the DESeq package.

### Flow cytometry for FCRL6 in PLHIV PBMCs

The PBMCs were cultured in RPMI 1640 medium (Gibco, Thermo Fisher Scientific, Waltham, MA, USA) supplemented with 10% human pool serum overnight and the stained for the following markers: LIVE/DEAD fixable viability dye viakrome (Beckman Colter - Brea, CA, USA), CD3 (BV605 - Biolegend - San Diego, CA, USA), CD4 (BUV395 - Biolegend), CD8 (BV785 - Biolegend), CD14 (AF700 - Biolegend), CD19 (PE-Cy5 - Biolegend), CD45 (Krome Orange - Beckman Colter - Brea, CA, USA), CD56 (BUV737 - Biolegend), TCRγδ (BV421 - Biolegend), and FCRL6 (APC - Biolegend). Antibodies were diluted in a total staining volume of 50 μL, which included Brilliant Stain Buffer Plus (BD Biosciences, Franklin Lakes, NJ, USA) and PBS supplemented with 2% bovine serum albumin (BSA- Sigma Aldrich, USA) and 2 mM EDTA (Gibco) (PBS-F) as the staining/washing buffer. For FcRL6-APC, a fluorescence minus one (FMO) control was included to define the positive populations. Staining was performed for 30 min at 4 °C in the dark, followed by washing with PBS-F + EDTA. Data acquisition was carried out using the CytoFLEX flow cytometer (Beckman Colter, Brea, CA, USA) (Supplementary Fig. S8A).

### Reporting summary

Further information on research design is available in the Nature Portfolio Reporting Summary linked to this article.

## Data availability

The raw data used in this study were previously published by Botey-Bataller et al.[39] and have been deposited in the Radboud Data Repository (https://doi.org/10.34973/p96d-kz55). These raw data are

available under restricted access to comply with the European Union General Data Protection Regulation for the protection of privacy-sensitive data. Access can be obtained by applications via the Radboud Data Repository data access portal. The timeframe for response to requests is within 4 weeks.

The processed data, including GWAS summary statistics, are available at Zenodo (https://doi.org/10.5281/zenodo.12580771). The association analyses results using immune cell counts, gene expression, plasma protein abundance, DNA methylation data generated in this study are provided in the Supplementary Information/Source Data file (Supplementary data 2–7 and Supplementary data 9–14).

## Code availability
Custom codes for the main analysis used in this study have been deposited on GitHub at https://github.com/CiiM-Bioinformatics-group/2000HIV_CMV. The code is also available via Zenodo (https://doi.org/10.5281/zenodo.17791464).

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

## Acknowledgements

We thank all the volunteers of the 2000HIV study. This study was supported by ViiV Healthcare (A18-1052); European Research Council (ERC) starting grant 948207 (Y.L.); a grant from the Lower Saxony Center for AI and Causal Methods in Medicine (CAIMed, Y.L.); the Deutsche Forschungsgemeinschaft (DFG) Fund (390874280) through Excellence Cluster RESIST (EXC 2155, Y.L.); Radboud University Medical Center Hypatia Grant (2018) (Y.L.); the European Union's Horizon 2020 research and innovation program under Marie Skłodowska-Curie grant agreement number 955321 (Y.L. and J.B.B.); the Lower Saxony MWK Sprung Fund (19777006, C.J.X.); DFG Fund 97673685 (C.J.X.); Deutsche Forschungsgemeinschaft (DFG) Fund (497673685, C.J.X.); Guangdong Basic and Applied Basic Research Foundation (2026A1515030018, Z.Z.); Singh Chhatwal Fellowships (2022, Z.Z.); and an ERC Advanced Grant (European Union's Horizon 2020 research and innovation program, grant agreement no. 833247, M.G.N.).

## Author contributions

Y.L., A.J.A.M.V., M.G.N., and C.J.X. conceived and supervised the research in the study and contributed equally to senior authorship. N.N. led the analyses, performed cytokine, cell count, transcriptomic, proteomic, and metabolomic data analysis, and wrote the initial draft.

 

Z.Z. performed GWAS analysis and wrote the corresponding text. X.J. performed methylation analysis and wrote the corresponding text. N.v.U. performed MR, xQTL analysis, and latent factor feature enrichment analysis, and wrote the corresponding text. V.M., J.C.S., and A.L.G. collected samples and generated and preprocessed omics data. L.Z. and J.B.B. assisted data analysis related to cell proportion and metabolomics, respectively. M.B., W.A.J.W.V., L.E., A.V., A.L., J.E.S., M.A.H.B., and L.A.B.J. collected samples and generated omics data.VM also did the MHC genetic analysis. All authors contribute to the writing and reviewing of the manuscript and approve the final version.

## Funding

## Competing interests
The authors declare no competing interests.

## Additional information

[1]Department of Computational Biology for Individualised Infection Medicine, Centre for Individualised Infection Medicine, a joint venture between the Hannover Medical School (MHH) and the Helmholtz Centre for Infection Research(HZI), Hannover, Germany. [2]TWINCORE, Centre for Experimental and Clinical Infection Research, a joint venture between the Hannover Medical School (MHH) and the Helmholtz Centre for Infection Research (HZI), Hannover, Germany. [3]Center for Cell Lineage Atlas, Guangzhou Institutes of Biomedicine and Health, Chinese Academy of Sciences, Guangzhou, China. [4]Department of Internal Medicine and Radboudumc Community for Infectious Diseases (RCI), Radboud University Medical Center, Nijmegen, the Netherlands. [5]Department of Internal Medicine, OLVG Hospital, Amsterdam, the Netherlands. [6]Department of Internal Medicine, Erasmus Medical Center, Rotterdam, the Netherlands. [7]Department of Medical Microbiology and Infectious diseases, Erasmus Medical Center, Rotterdam, the Netherlands. [8]Department of Internal Medicine, Elisabeth-Tweesteden Hospital, Tilburg, the Netherlands. [9]These authors contributed equally: Nhan Nguyen, Zhenhua Zhang, Xun Jiang, Nienke van Unen. ✉e-mail: yang.li@helmholtz-hzi.de

