## [Peer Review file · Nature Communications]

Molecular Signatures and Causal Factors Underlying Latent Cytomegalovirus Infection Among People Living With HIV (PLHIV)

Corresponding Author: Ms Nienke van Unen

Version 0:

Reviewer comments:

Reviewer #1

(Remarks to the Author)

This study comprehensively investigated the CMV-induced molecular phenotypes in a PLHIV cohort. The authors identified several immune signatures related to CMV seropositivity, especially FCRL6. There are several concerns listed as below.

Major points:

1. The first big concern is the strong imbalance of the CMV+ and CMV- data size. The CMV+ group has much more samples than CMV- group in both discovery and validation cohort. How did the authors make sure the results are not biased by the imbalance?

For Figure 1B, for example, within the CMV+ group, there might be also heterogeneity, if the authors take 500 random CMV+ samples and compare with other 500 random CMV+ samples, how does the effect size distribution look?

If the authors try to downsample the CMV+ group to the same size of CMV- group and re-perform the analysis, would the results be still similar?

2. How much of the CMV + vs CMV- results are specific to PLHIV?

Do the healthy CMV+ individuals also have the similar immune activation as observed in the PLHIV? How do these molecular phenotypes connect to the HIV complications?

3. Could the authors provide quality control information for the bulk RNA-seq data? For example, a PCA plot of all the samples.

4. Could the author perform colocalization analysis to see if the GWAS and eQTL are sharing causal variants? Please include the LD information in Fig 4C. Is CHRN4 the gene that mediates this protective effect? More biological insights (such as tissue/cell type expression patterns) might be added in addition to the current discussion.

5. Please provide the statistical test results for Fig 4F and G.

6. Could the authors provide the complete results of pQTL and cQTL analysis instead of just showing 1 example in Figure 4F, G, J)?

A comprehensive multiple QTLs integration analysis should be performed and could serve as an independent figure itself.

7. Is there pQTL/eQTL information for FCRL6?

Minor points:

1. Could the authors arrange the supp Figures in the chronological order?

2. Line 243, il1ra should be IL1RA, the first letters should be capital.

3. In Fig2I, what is the correlation coefficient between methylation and gene expression?

4. Line249, FigS1H is wrong, there is no FigS1H.

5. Figure4 A bottom got covered by 4C.

6. Line 347, where is the Figure 5E?

7. The row names in Fig1C are not easy to read, maybe the authors could try to remove the 'pbmc_' and also somehow rearrange the order?

8. Line257, is the validation using p value instead of adjusted p value?

(Remarks on code availability)

Reviewer #2

(Remarks to the Author)

This study investigates the immunological, molecular, epigenetic and genetic impacts of cytomegalovirus (CMV) seropositivity in people living with HIV (PLHIV). CMV+ individuals exhibited heightened immune activation following PBMC exposure to CMV antigen. CMV seropositivity was also associated with significant DNA methylation changes affecting immune-related gene expression, highlighting eight consistent markers, including FCRL6, which emerged as a potential marker for immune activation. GWAS revealed a CMV-seroprevalence locus on chromosome 15 (rs7180928-G) linked to cytokine production and plasma protein abundance. Mendelian randomization analyses suggested that increased FCRL6 expression is genetically driven and associated with latent CMV infection. These findings underscore the role of CMV in immune modulation in PLHIV and suggest broader implications for other immunocompromised populations.

The study has generated significant results with sound methodology. Here are a couple of recommendations for the paper.

1. Methylation measured on bulk cells. Could the authors perform a deconvolution analysis and show if the specific cells that are infected with CMV show the overall pattern observed.

2. It has been demonstrated that the non-classical HLA molecule, HLA-E, is responsible for CMV control. The results from the manuscript indicate an NK related function, which is interesting since HLA-E is linked to NK cell function. Could the authors examine the HLA-E region specifically to determine a direct methylation/transcriptome/GWAS effect on CMV. This will identify a new pathway and future validate the data generated in this manuscript with the previously published HLA-E gene linked to CMV.

(Remarks on code availability)

Reviewer #3

(Remarks to the Author)

This is a comprehensive multi-omic analysis of the relationship of CMV seropositivity (latent CMV infection) with the immune system in people with HIV, as part of the 2000HIV project. The authors have observed profound effects of having CMV on gene expression, cytokine production, plasma proteome and metabolome, and DNA methylation in the peripheral blood and in peripheral blood cells, using a discovery and a validation cohort approach. While some of the differences are expected from previous literature, this study is notable for its very meticulous approach, the large numbers of individuals analyzed by each method, and the connections among the different assays, which demonstrate regulation at several levels – for instance, CMV is associated with DNA de-methylation of FCRL6 (consistent with its upregulated gene expression and protein abundance) in CMV-seropositive individuals with HIV. Finally, the authors use GWAS and mendelian randomization analysis to posit that the FCRL6 gene is causally linked to CMV infection, although how the authors think FCRL6 leads to CMV susceptibility is not well described.

Major Concerns:

- The data in Figure 1C, showing increased cytokines produced by PBMCs of CMV-seropositive donors after CMV pp65 stimulation for 24h seems to be a bit confusing to me. The authors don't really justify why CMV pp65 peptide (an internal matrix protein) should elicit innate cytokines directly. And an obvious alternative - that pp65 protein is processed and presented to CMV-reactive T cells in the culture, which then release cytokines (such as TNF or IFN γ) to activate the innate cells – is seemingly not considered. Indeed, this may explain the reduced TNF in the validation cohort after CMV pp65 stimulation (the darkest blue cell): innate cells are taking up the TNF.
- The authors go to a lot of effort to investigate the role of CMV on FCRL6 (and vice versa), but there is no data on which cells are expressing it in vivo, and whether cellular FCRL6 levels are different in CMV seropositive and seronegative individuals. They say from the literature that it is expressed on NK cells and T cells – could the authors stain PBMCs from CMV disparate individuals and show a difference in FCRL6 surface expression?
- Could the authors discuss why none of the SNPs (instrumental variables) shown in Figure 5C (causal for gene expression) overlap with the SNPs in Figure 5E (causal for protein expression), and vice versa. Is this an expected finding or not?

Minor Concerns:

- Please refrain from using terms like “HIV-infected” or “non-HIV-infected”. Just use people (living) with HIV or people without HIV. The term “patients” is also discouraged.
- Table S7 only seems to contain 700 DEGs (FDR <0.05), not 807 as stated in the text (line 257).
- Figure 5 panel E is not labeled within Figure 5.
- Line 587 in the Methods says “(show figures)” ???
- Figure 2B legend (lines 769-771) the x-axis and y-axis designations are reversed from what is shown in Figure 2B.
- Figure 2F legend (line 780) says “MOMER” instead of “HOMER”.

(Remarks on code availability)

Reviewer #4

(Remarks to the Author)

This is an interesting manuscript that assesses CMV serostatus in relation to a large study of 'omics data in a cohort of people living with HIV. The work develops hypotheses related to the impact of CMV on the immune system.

Points

- The clinical cohort has a very high rate of CMV seroprevalance and this may reflect demographic factors related to infection risk. The relevance of the data to the general population is thus not clear and this should be stated as a limitation.
 - I find the manuscript to be rather poorly written for the scientific reader. I would advise a considerable rewriting that addresses the cohort structure, experiments that were done in each section and the specific outcomes. The abstract also needs to be succinct and clear
 - The cytokine functional assessment is in relation to the short term response to challenge with a CMV protein. CMV modulates the immune systems in substantial ways and as such this differential response is to be expected.
 - the work uses bulk transcriptomics and the increase in many cytotoxic transcripts is not surprising.
 - CMV induces major changes in the immune cell repertoire - has this been corrected for in relation to the relative prevalence of genes and methylation profiles associated with this transition?
 - The work also seeks to assess genomic risk factors related to CMV infection. There has been prior work in this regard and the incremental advance here seems modest. The potentially unusual features of the residual CMV-seronegative donors within this cohort should be discussed.
- The FCRL6 association is interesting and the most important feature of the manuscript. The magnitude of this effect, and as such its potential use in assessment of CMV status within individual donors, could be expanded.

(Remarks on code availability)

NA

Version 1:

Reviewer comments:

Reviewer #1

(Remarks to the Author)

I appreciate the authors' efforts during the revision. The manuscript has been improved significantly. Here are a few points I hope could be elaborated further.

1)Major point 1.

I appreciate the downsampling analysis and suppFig S3,S4,S5. Just out of curiosity, how was the number of downsample size '86' determined? Any intuition behind and maybe included in the methods section?

A minor suggestion for FigS4-S5: Maybe the FCRL6 can be highlighted using a different color or shape for the point. Also the 'validate_DEGs/DEPs yes/no' labels only need to show once for the whole plot instead of in each subplots.

2)Major point 4.

Could the authors provide PPH1 and PPH2 of the coloc analysis as well?

Could the author also add a box plot for CHRN4 eQTL? Please also report the accurate nominal p value of the CHRN4 eQTL for rs7180928 instead of 'P<0.05'.

I appreciate the new locus zoom plot with LD information. But the Figure legend for Fig4C (line) seems to be not updated (there is no orange dots anymore). Now the eQTL and GWAS signals are not easy to distinguish (dark blue vs light blue?). Could the authors try to improve it?

Maybe another more straightforward way is to plot them separately but just stack the GWAS on top of the eQTL and use the same gene annotation panel.

It would be better to have 2 y axis (left and right) showing for both $-\log_{10}p$ value of eQTL and of GWAS, if you still overlay the 2 plots.

According to the original Figure4C, the lead eQTL for CHRNA4 is rs11629784. What is the p value of this one in the GWAS?

I appreciate the author's effort of finding CHRNA4 related studies. However, I am still not convinced this is the most likely mediating gene. According to GTEX data, rs7180928 has eQTL effect for CRABP1 ($p=0.0000041$) and for DNAJA4 ($p=0.000016$) (see <https://gtexportal.org/home/snp/rs7180928>), are they not significant or not detectable in your analysis? There are quite a bunch of genes (e.g. SKIC8, DNAJA4, CRABP1, IREB2, HYKK, PSMA4, CHRNA3, CHRNA5) near the GWAS peak in this locus. I assume the authors have full eQTL summary statistics for all these genes. I guess some of these genes in addition to CHRNA4 should also have eQTL signal. I would suggest the authors to run coloc between each of these eQTLs and GWAS, then to check if the CHRNA4 eQTL really has the highest PPH4. This analysis should be able to make sure that CHRNA4 is the most likely mediating gene.

3)Major point 5.

Please add the KIR2DS4 and IL22 to the y axis labels of the Fig4F,G. Also for this kind of QTL box plots, I think the authors should have a p value for each, instead of comparing different genotypes 3 times. Could you label that p value from the QTL analysis? Please also report the test method in the figure legend.

All the other points were well addressed, thanks.

Please also update the code for the new analysis on github.

(Remarks on code availability)

Reviewer #2

(Remarks to the Author)

I am satisfied with the changes the authors have made

(Remarks on code availability)

The code has been checked and was functional and appropriate

Reviewer #3

(Remarks to the Author)

The authors have done an outstanding job responding in great detail to the reviewers' comments, including my own. The manuscript is much improved and I don't have any major concerns that would preclude acceptance.

Minor concerns:

-There are some typos (words without spaces between, etc) and other English language issues, particular on lines:

52 ("populationsassociated")

52 ("seropositivity.This")

57 ("PLHIV. ost")

103 ("to90%")

240 ("cytokines PBMCs")

294-295 ("also occur HIV-specific")

320 ("not unsurprising") [I think you mean not surprising here]

-In addition, many of the references appear to be messed up, with only initials given instead of names: Refs 4, 6, 14, 16, 17, 19, 20, 21, 22, 24, 25, 26, 27, 28, 30, 31, 32, 34, 35, 36, 43, 44, 46, 48, 49, 53, 54, 55, 57, 58, and 72.

(Remarks on code availability)

Reviewer #4

(Remarks to the Author)

This is a revision.

The authors have answered the reviewers comments very well. They have maximised the power of the study cohort.

The demonstration of increased levels of FCRL6 on several immune cells is important.

There are spelling mistakes in the abstract which is surprising.

(Remarks on code availability)

Reviewer #1 (Remarks to the Author):

This study comprehensively investigated the CMV-induced molecular phenotypes in a PLHIV cohort. The authors identified several immune signatures related to CMV seropositivity, especially *FRCL6*. There are several concerns listed as below.

We thank Reviewer #1 for recognising the comprehensive nature of our multi-omic investigation into CMV-induced molecular phenotypes in PLHIV and for highlighting *FCRL6* as a key immune signature.

Major points:

1.1. The first big concern is the strong imbalance of the CMV+ and CMV- data size. The CMV+ group has much more samples than CMV- group in both discovery and validation cohort. How did the authors make sure the results are not biased by the imbalance?

For Figure1B, for example, within the CMV+ group, there might be also heterogeneity, if the authors take 500 random CMV+ samples and compare with other 500 random CMV+ samples, how does the effect size distribution look?

If the authors try to downsample the CMV+ group to the same size of CMV- group and re-perform the analysis, would the results be still similar?

We thank the reviewer for this important comment regarding the imbalance between CMV+ and CMV- groups.

We agree a downsampling analysis would be a good way to ensure that our findings are not biased by the larger sample size in the CMV+ group. Therefore, we conducted the association analysis 100 times, each time randomly selecting **86 CMV+** individuals to match the **86 CMV-** individuals in the discovery cohort. We then compared the effect sizes from these 100 downsampled analyses to those obtained from the full discovery cohort.

The results were highly consistent. The associations retained the same direction of effect across downsampling iterations.

- **DNA methylation** (Figure R-1): more than 85% of the top associations were replicated in the downsampled datasets.
- **Transcriptomics** (Figure R-2): on average, around 71.49% significant genes (FDR < 0.05) were replicated in the downsampled dataset. Importantly, the gene of interest (*FRCL6*) was significant in all the downsampling analysis.
- **Proteomics** (Figure R-3): on average, around 80.13% significant proteins (FDR < 0.05) were replicated in the downsampled dataset. Importantly, the protein of interest (*FRCL6*) was also significant in all the downsampling analysis.

Figure R-1. Correlation between effect sizes from the full DNA methylation analysis (x-axis) and the downsampling analyses (y-axis). The darker colour indicates if the CpG site was also significant in the respective downsampling analysis (FDR < 0.05). The percentages of significant CpG sites (FDR < 0.05) replicated in the downsampling iterations are in the range between 85% - 98%. *This figure has been included in the revised manuscript as Supplementary Fig. S3.*

Figure R-2. The concordance plots for re-performing the bulk RNAseq analysis across 100 downsampling iterations in the discovery cohort. The percentages of significant genes (FDR < 0.05) replicated in the downsampling iterations are in the range between 48.89% - 86.55% with mean = 71.49%. The x-axis is the effect size from original analysis. The y-axis is the effect size from downsampling analysis. *This figure has been included in the revised manuscript as Supplementary Fig. S4.*

Figure R-3. The concordance plots for re-performing the proteomics analysis across 100 downsampling iterations in the discovery cohort. The percentages of significant proteins (FDR < 0.05) replicated in the downsampling iterations are in the range between 44.74% - 100.00% with mean = 80.13%. The x-axis is the effect size from original analysis. The y-axis is the effect size from downsampling analysis. *This figure has been included in the revised manuscript as Supplementary Fig. S5.*

These findings demonstrate that the key results are robust and not driven by group size imbalance. We included the downsampling analyses methods and outcomes in our revised manuscript as followed:

Page 9, line 219: “and more than 85% could be replicated in the downsampled analysis (Supplementary Fig. S3).”

Page 10, line 262: “A downsampling analysis was performed 100 times with random sample selections to evaluate the potential bias due to the imbalance between the CMV+ and CMV- patient group size. On average, around 71.49% of 1442 DEGs in the discovery cohort were replicated in the downsampling analysis (FDR < 0.05) (Supplementary Fig. S4).”

Page 17, line 499: “To evaluate the potential bias due to the imbalance between the CMV+ and CMV- group size, we performed a downsampling analysis with random sample selections 100 times. On average, around 80.13% significant proteins in the discovery cohort were replicated in the downsampling analysis (FDR < 0.05) (Supplementary Fig. S5)”

And new methodology (Methods ‘Downsampling Analysis’, **page 25, lines 760-767**):

“To assess whether our larger CMV+ sample introduced bias, we repeated the seropositivity association analysis 100 times on downsampled datasets for DNA methylation, gene expression, and protein abundance. In each iteration, we randomly selected 86 CMV+ individuals to match the 86 CMV- individuals in the discovery cohort (n = 172). We then compared the effect sizes from these 100 downsampled analyses to those from the original full-cohort analysis (n = 1553). Finally, we calculated the proportion of molecules (FDR < 0.05) whose associations were consistently replicated across the downsampling iterations.”

1.2. How much of the CMV + vs CMV- results are specific to PLHIV?

Do the healthy CMV+ individuals also have the similar immune activation as observed in the PLHIV?
How do these molecular phenotypes connect to the HIV complications?

Thank you for your questions. We have answered them in subparts below.

1.2a. How much of the CMV + vs CMV- results are specific to PLHIV?

In addition to the previously reported 98.7% concordance of CMV-associated DNA methylation changes with an independent healthy cohort, we have now extended our analysis to protein abundance in the Dutch young healthy (300BCG) cohort (DOI: [10.1172/JCI133935](https://doi.org/10.1172/JCI133935)). Out of the 273 proteins showing $P < 0.05$ in our PLHIV analysis, nine were included in the healthy cohort. All of these had concordant direction of effect, and six were also nominally significant. This high level of replication at the protein level, together with our initial methylation concordance, indicates that CMV-driven molecular alterations are robust and occur independently of HIV infection.

We have revised our manuscript and added the new **Results** on **page 11, lines 294-300**:

“To determine whether the CMV-associated proteomic changes we observed in PLHIV also occur HIV-specific, we analysed the 300BCG healthy cohort (CMV+ = 74, CMV- = 241), which measured 73 proteins using an inflammatory Olink panel³⁸. Nine proteins out of the 273 significant proteins ($P < 0.05$) were shared between datasets, of which three were nominally significant in both cohorts and all with concordant direction of effect (Supplementary Fig. S6D). Though the sample size in the healthy cohort is relatively small, this result indicates that the CMV-driven protein alterations appear to be independent of HIV status.”

Figure R-4. Comparison of effect sizes between nominally significant proteins ($P < 0.05$) between CMV+ and CMV- in PLHIV and matching proteins in a healthy cohort. *This figure has been included in the revised manuscript as Supplementary Fig. S6D.*

As well as new Methods on **page 24, lines 732-735**:

“For a comparative analysis in healthy individuals, we used Olink data from the Dutch 300BCG study³⁸ (CMV+ = 74, CMV- = 241). Out of the 273 significantly differentially expressed proteins in the PLHIV discovery set, nine were also measured in the inflammatory panel used for this cohort. For these nine proteins we fitted a linear model, correcting for age and sex effects.”

1.2b. Do the healthy CMV+ individuals also have the similar immune activation as observed in the PLHIV?

We appreciate the reviewer’s insightful question. While cytokine response data are available for the healthy cohort mentioned in the previous question, these responses were not measured following CMV-specific stimulation. As such, a direct comparison to the CMV-specific immune activation observed in PLHIV is not currently feasible. We acknowledge this as a limitation and hope to explore this question in future studies with matched antigen-specific assays.

1.2c. How do these molecular phenotypes connect to the HIV complications?

We thank the reviewer for this question. Building on our recently published Multi-Omics Factor Analysis (MOFA) analysis in the European discovery subset ($n = 996$; integrating gene expression, plasma proteins, metabolites, cytokine responses and DNA methylation; DOI: [10.1038/s41591-025-03887-1](https://doi.org/10.1038/s41591-025-03887-1)), we performed an over-representation analysis (ORA) between the CMV-associated molecular traits identified here and the trait loadings that define each latent factor (LF). We excluded DNA methylation as the direction of effect can be more ambiguous than the other layers.

For each LF significantly associated with a clinical phenotype, loadings were given a “risk” or “protective” label:

- if the LF correlated positively with the phenotype, features with positive loadings were classified as risk;
- if the LF correlated negatively, features with negative loadings were classified as risk.

Out of four clinical phenotype-linked LFs, three were significantly enriched for genes and proteins up or down-regulated in CMV+ individuals (Figure R-5): Factor 8, correlated with **hypertension and prior myocardial infarction**, (ii) Factor 11, associated with **chronic obstructive pulmonary disease (COPD)**, (iii) Factor 20, associated with the **rapid-progressor** phenotype in HIV infection.

We added these new results to a new **Results** section titled ‘CMV-responsive signals converge on cardiopulmonary and HIV-progression factors’ on **pages 11-12, lines 301-330**:

“Building on our previously published Multi-omics Factor Integration (MOFA) model of the European discovery subset ($n = 996$)³⁹, we asked whether CMV-associated molecular traits map onto the latent pathways already linked to comorbidities in PLHIV. We tested for over-representation of the CMV+ and CMV- differentially expressed molecules among the highest-weight features of the four disease-associated latent factors (LF6 plaque, LF8 hypertension & myocardial infarction, LF11 COPD, LF20 rapid progressors). We excluded DNA methylation from this analysis due to the direction of effect being more ambiguous than the other layers. Loadings were classified as risk or protective according to the direction of each factor’s correlation with its disease phenotype.

Three of the four clinical latent factors (LF8, LF11 and LF20) show a clear CMV imprint (FDR<0.05 across several top-weight cut-offs; Supplementary Fig. S7). LF11 is particularly striking. This factor, originally linked to both COPD risk and higher “latest CD8” counts, shows a significant positive correlation with CMV serostatus and a linear relationship with circulating CMV-IgG titres (Supplementary Fig. S7A,B). Its risk weights are enriched for CMV-upregulated genes and plasma proteins, whereas the protective tail is over-represented by CMV-down-regulated transcripts (Supplementary Fig. S7C). This polarity suggests that latent CMV infection drives transcriptional programmes that feed into cytotoxic profile captured by LF11, thereby predisposing to COPD pathogenesis. By contrast, transcripts reserved in CMV- individuals may buffer against excessive CD8+ activation and lung inflammation. This is not unsurprising, as COPD pathogenesis is tightly coupled to excessive CD8+ T-cell driven inflammation⁴⁰. In addition, while the impact of CMV co-infection on COPD has not yet been explored in PLHIV, chronic CMV infection itself has been associated with COPD in HIV-uninfected populations⁴¹.

For LF8, the factor linked to cardiovascular diseases, the risk features were enriched for CMV-upregulated protein expression (Supplementary Fig. S7D). This agrees with earlier reports linking CMV co-infection to cardiovascular comorbidity in PLHIV⁵. LF20, capturing rapid-progressor HIV phenotype, showed a similar CMV+ enrichment in its risk weights for gene expression signatures (Supplementary Fig. S7E), echoing evidence that CMV accelerates HIV disease progression⁴². All in all, the patterns shown here suggest that CMV infection amplifies molecular programmes promoting heart and lung inflammatory diseases and HIV disease progression in PLHIV.”

Figure R-5. Results of the enrichment of the CMV-associated traits for three latent factors. We tested for enrichment iteratively from 5% until 95% of the top features for each LF. Only dots passing multiple testing corrections are colored ($FDR < 0.05$). *These figures have been included in the revised manuscript as Supplementary Fig. S7.*

As well as new methodology (Methods ‘Cross-referencing CMV-responsive signatures with published MOFA latent factors’, **page 25, lines 768-787**):

“To assess the role of CMV-associated molecules, we utilised a previously published Multi-Omics Factor Analysis (MOFA) model of the European discovery subset (n = 996) which integrated matched whole-blood RNA-seq, plasma proteomics (Olink), untargeted metabolomics, and ex-vivo cytokine responses³⁹. The complete factor-feature weight matrix from that study served as the reference for all enrichment analyses described below; no additional factor training or tuning was performed in the present work.

Within four latent factors linked to clinical phenotypes, loadings were ranked separately by absolute positive and negative weights. Based on the direction of the correlation to the clinical phenotype, we labelled loadings as risk or protective for the respective disease. E.g. if the LF correlated positively with the phenotype, positive weights were deemed risk, if the LF correlated negatively, negative weights were deemed risk.

Eleven percentile cut-offs (5%-95% in 10% steps) defined nested top-N feature lists for both directions. At each cut-off we performed one-tailed hypergeometric tests comparing the selected features against the corresponding CMV discovery lists, restricted to the view-specific universe. Four tests were evaluated per percentile (risk-up, risk-down, protective-up, protective-down). Benjamini-Hochberg correction was applied per LF across all percentiles and test combinations. Significant enrichments (adjusted P < 0.05) were depicted as dot plots: point size represents percent overlap, colour encodes $-\log_{10}(\text{padj})$, facets separate risk vs protective weights and CMV+/CMV-discovery sets.”

1.3. Could the authors provide quality control information for the bulk RNA-seq data? For example, a PCA plot of all the samples.

Thank you for requesting additional QC details for our bulk RNA-seq data. We have revised the manuscript (Methods ‘Gene expression measurement and analysis’, **page 22, lines 656-669**) to include the full QC and filtering workflow as follows:

“Bulk RNA-seq of PBMCs was performed on Illumina platforms (>30 million reads per sample). Sequencing reads were aligned to the human reference genome (NCBI build 38) using STAR. Key QC metrics (read quality, duplication rates, alignment rates, gene-body coverage, etc.) were aggregated via MultiQC, and all samples passed standard thresholds. We then applied the following sample-level filters to the initial set of 1,899 libraries: (i) excluding 2 libraries per Nijmegen team decision, (ii) removed 1 library with no uniquely mapped reads, (iii) discarded 3 duplicate-donor libraries, retaining the replicate with higher read depth, (iv) excluded 29 libraries with <5 million uniquely mapped reads, (v) excluded 7 libraries with discordant reported sex versus Y-chromosome read counts. This resulted in 1,857 final samples (Supplementary Fig. S10). Importantly, no samples were removed as PCA outliers, in order to retain biologically relevant variation (Supplementary Fig. S10). For gene-level QC, 58,347 transcripts were detected, of which those with ≥ 10 counts in the smallest

comparison group were retained, yielding 16,000 to 20,000 genes per analysis. The reads underwent rlog transformation and normalisation using DEseq2 package⁶⁸.”

We have also added the requested PCA plot of all 1,857 samples (Figure R-5). We hope these additions fully address your comment.

Figure R-5. PCA was performed on normalized expression values of all genes in the discovery cohort. The first principal component (PC1) explains 22% of the variance and the second principal component (PC2) explains 6.8% of the variance. Each point represents a single individual, color-coded by clinical condition: HIV-infected progressors (HIV; red), elite controllers with transient CMV reactivation (EC_TRANSIENT; cyan), rapid progressors (RP; orange), and elite controllers with persistent CMV reactivation (EC_PERSISTENT; dark red). Shaded ellipses denote the 95% confidence interval for each group. *This figure has been included in the revised manuscript as Supplementary Fig. S10.*

1.4. Could the author perform colollization analysis to see if the GWAS and eQTL are sharing causal variants? Please include the LD information in Fig4C. Is CHRN4 the gene that mediates this protective effect? More biological insights (such as tissue/cell type expression patterns) might be added in addition to the current discussion.

Thank you for your questions. We have answered them in subparts below.

1.4a. Could the author perform colollization analysis to see if the GWAS and eQTL are sharing causal variants?

We agree that the colocalization analysis will confirm the underlying shared causal variants. Following the reviewer’s suggestion, we have performed colocalization analysis between eQTL of CHRN4 and GWAS of CMV. The coloc returns a PP4 = 0.34 and PP3 = 0.23. The combination of these two posterior p-values suggest a correlation between CHRN4 expression and CMV+/-, while they likely share the same causal variant. However, limited by the sample size, we did not reach the empirical threshold of PP4 >= 0.5. Therefore, we move forward to perform a one sample Mendelian randomization (MR) analysis. The result suggested a potential causal relationship between CHRN4 expression and CMV-seronegative outcome in our cohort (Figure R-6). Collectively, the colocalization and MR analysis support the role of CHRN4 in controlling CMV resistance in PLHIV. These results have been added to the revised manuscript (page 14, lines 379-386):

“To further confirm the role of CHRN4 in CMV infection in PLHIV, we performed colocalization and MR analyses to verify if the gene mediates PLHIV’s resistance to CMV. The colocalization resulted in a PP4 = 0.34 and PP3 = 0.23 (Bayesian factor analysis), suggesting a gentle but potential shared locus between CHRN4 expression and CMV-seronegative outcome. Subsequently, we performed a one-sample MR analysis to further support the observation using eQTL and GWAS summary statistics. The result indeed supports the colocalization analysis (Supplementary Fig. S9C). However, due limited expression of CHRN4 in whole blood, future work is required to confirm these observations.”

Figure R-6. MR analysis between CHRN4 expression and PLHIV resistance to CMV. *This figure has been included in the revised manuscript as Supplementary Fig. S9C.*

1.4b. Please include the LD information in Fig4C.

We thank the reviewer for the suggestion to add LD information in Fig. 4C. We feel this information will definitely improve the figure. Due to the new layout, we have adjusted the figure by splitting the two summary statistics, updating the figure legend as well. The new figure can be found below (Figure R-7).

Figure R-7. New panel C for Figure 4, including the LD information.

1.4c. Is CHRNA4 the gene that mediates this protective effect? More biological insights (such as tissue/cell type expression patterns) might be added in addition to the current discussion.

We thank the reviewer’s suggestion to dive into the locus to gain more biological insights. From the results above, we have concluded that CHRNA4 mediates the protective effect. However, the CHRNA4 gene is specifically expressed in the testis (Figure R-8). We did not observe expression patterns between CMV+ and CMV- due to limited expression of CHRNA4 in whole blood. Previous studies have suggested the role of CHRNA4 in lung function (e.g., lung cancer and COPD, 10.3389/fonc.2020.571167, 10.1038/s41467-021-26637-6), and addiction to nicotine ([10.1038/npp.2010.95](https://doi.org/10.1038/npp.2010.95), [10.1093/hmg/ddr498](https://doi.org/10.1093/hmg/ddr498), [10.1038/ng.571](https://doi.org/10.1038/ng.571), [10.1038/mp.2013.158](https://doi.org/10.1038/mp.2013.158)) etc, while there is limited studies clearly state the relation between CMV infection and CHRNA4 expression. Of note, Lunardi et al have suggested the response of CHRNA4 in fibroblasts upon the treatment of anti-hCMV antibodies (Table 2, 10.1371/journal.pmed.0030002), suggesting the potential role of CHRNA4 in CMV infection, which has been discussed in the manuscript (page 17, line 498).

Figure R-8. Violin plots show the tissue expression of CHRNA4 across tissue from the GTEx portal.

1.5. Please provide the statistical test results for Fig4F and G.

Thank you for this suggestion. We have now added the statistical test results in Figure 4F and G, as shown below in Figure R-9.

Figure R-9. New panels F (left) and G (right) for Fig. 4, showing the statistical tests between the violin plots.

1.6. Could the authors provide the complete results of pQTL and cQTL analysis instead of just showing 1 example in Figure 4F,G,J)? A comprehensive multiple QTLs integration analysis should be performed and could serve as an independent figure itself.

We appreciate the reviewer's suggestion. The full results of the pQTL and cQTL analyses (currently filtered at the threshold of $P < 1 \times 10^{-5}$), as well as eQTL, are publicly available on our website. These analyses were conducted as part of a comprehensive multiple QTL integration analysis by our group, and the corresponding manuscript is currently online at Nature Medicine.

To indicate that our *FCRL6* QTL analysis is part of a larger comprehensive analysis, we have added the following text under the QTL mapping Methods on **page 27, line 830**:

"The full summary statistics can be found at https://lab-li.ciim-hannover.de/apps/hiv_xqtl_atlas/."

1.7. Is there pQTL/eQTL information for *FCRL6*?

Thank you for your question. Yes, in fact we used the eQTL and pQTL information for *FCRL6* in our **Mendelian Randomization (MR)** analysis shown in Figure 5, selecting significantly associated genetic variants as instrumental variables (IV). The IV for the *FCRL6* gene expression MR can be found in Figure 5C, and the ones for the *FCRL6* protein MR are shown in Figure 5E.

To indicate this more clearly, we have modified the MR results to mention that the IV are derived from QTL summary statistics on **page 14, line 405**: "Mendelian Randomization (MR) leverages genetic variants (SNPs) associated with *FCRL6* gene expression (eQTL) or protein abundance (protein

QTL; pQTL) as instrumental variables (IVs) to infer causal relationships between these exposures and CMV seropositivity.”, and by adding “(eQTL)” and “(pQTL)” to the Methods ‘Mendelian Randomization’ on **page 27**.

Minor points:

1. Could the authors arrange the supp Figures in the chronological order?

Thank you for this suggestion. We have now rearranged the Supplementary Figures to reflect the chronological order in which they are referenced in the manuscript, enhancing readability and clarity.

2. Line 243, il1ra should be Il1ra, the first letters should be capital.

Thank you for pointing this out. We have corrected il1ra to Il1ra as suggested in the mentioned line.

3. In Fig2I, what is the correlation coefficient between methylation and gene expression?

Thank you for your question. The correlation coefficient (r) between methylation and gene expression is -0.31. The details of the calculation are described in the methods section.

4. Line249, FigS1H is wrong, there is no FigS1H.

Thank you for pointing this out. It should be “Fig. 2H”. The manuscript has been updated accordingly.

5. Figure4 A bottom got covered by 4C.

Thank you for pointing this out. We have adjusted the panel positions to fix this issue.

6. Line 347, where is the Figure 5E?

Thank you for catching this. The label for Figure 5E was missing, and we have now added it in the revised manuscript.

7. The row names in Fig1C are not easy to read, maybe the authors could try to remove the ‘pbmc_’ and also somehow rearrange the order?

Thank you for the suggestion. To improve readability, we have removed the “pbmc_” from the figure (Figure R-9). Although the order does not follow an alphabetical pattern or their exposure duration, the order was arranged via a clustering calculation, so the cytokines were grouped based on the similarity in their sample profiles to each other. For instance, the current plot is able to show cytokines that were upregulated in CMV+ grouped near each other. Thus, the current order might help readers to navigate the plot and focus on the up/down-regulated pattern in cytokines.

Figure R-9. New panel C for Fig. 1.

8. Line257, is the validation using p value instead of adjusted p value?

Thank you for your question. In the submitted version, the validation was performed using the p-value, not the adjusted p-value. However during this revision, we updated this to FDR < 0.05 based on another reviewer's suggestion, resulting in 700 DEGs being validated in the validation cohort.

Reviewer #2 (Remarks to the Author):

This study investigates the immunological, molecular, epigenetic and genetic impacts of cytomegalovirus (CMV) seropositivity in people living with HIV (PLHIV). CMV+ individuals exhibited heightened immune activation following PBMC exposure to CMV antigen. CMV seropositivity was also associated with significant DNA methylation changes affecting immune-related gene expression, highlighting eight consistent markers, including FCRL6, which emerged as a potential marker for immune activation. GWAS revealed a CMV-seroprevalence locus on chromosome 15 (rs7180928-G) linked to cytokine production and plasma protein abundance. Mendelian randomization analyses suggested that increased FCRL6 expression is genetically driven and associated with latent CMV infection. These findings underscore the role of CMV in immune modulation in PLHIV and suggest broader implications for other immunocompromised populations.

The study has generated significant results with sound methodology. Here are a couple of recommendations for the paper.

We appreciate Reviewer #2's positive assessment of our study's robust methodology and significant findings across immunological, molecular, epigenetic, and genetic layers.

2.1. Methylation measured on bulk cells. Could the authors perform a deconvolution analysis and show if the specific cells that are infected with CMV show the overall pattern observed.

We thank the reviewer for this insightful suggestion. We fully agree that distinguishing DNA methylation patterns between CMV-infected and non-infected cell types would enhance the resolution of our analysis. However, the methylation data in our study were generated from whole blood samples. Given the latent nature of CMV infection and the absence of single-cell level annotation or definitive markers to identify infected cells, we currently lack the resolution to reliably distinguish between infected and non-infected cell populations. Moreover, to our knowledge, there are no established deconvolution methods capable of inferring CMV infection status at the cell-type level from bulk methylation data. Nevertheless, we consider this an important future direction and will explore it further as methodological advancements become available.

2.2. It has been demonstrated that the non-classical HLA molecule, HLA-E, is responsible for CMV control. The results from the manuscript indicate an NK related function, which is interesting since HLA-E is linked to NK cell function. Could the authors examine the HLA-E region specifically to determine a direct methylation/transcriptome/GWAS effect on CMV. This will identify a new pathway and future validate the data generated in this manuscript with the previously published HLA-E gene linked to CMV.

We thank the reviewer for this insightful suggestion regarding the role of HLA-E in CMV control.

To explore potential genetic regulatory effects, we examined the HLA-E genomic region in our CMV **GWAS** and found that the top SNP (rs2021368, $p = 4.25e-05$) lies nearby HLA-E (Figure R-10). This indicates a suggestive association.

Figure R-10. Locus plot of the HLA-E locus from the CMV GWAS. *This figure has been included in the revised manuscript as Supplementary Fig. S9F.*

Our **gene expression** analysis revealed that *HLA-E* expression in PBMCs showed a nominally significant association with CMV seropositivity ($p = 0.03$, Supplementary Table S6), making it the strongest association among the non-classical HLA genes. However, the effect size remains modest, suggesting that while HLA-E expression by peripheral blood mononuclear cells (PBMCs) may play a role, its contribution to CMV serostatus is not the primary driver.

In contrast to the GWAS and transcriptome, plasma **protein** levels of circulating HLA-E did **not** show any significant associations with CMV serostatus ($p = 0.14$, Supplementary Table S9).

Lastly, we analyzed whole blood **DNA methylation** data for associations with CMV seropositivity across non-classical HLA genes. Here we also did **not** observe a strong association between CMV serostatus and methylation at HLA-E. However, we did identify a notable association at **HLA-F/HLA-F-AS1** (Table R1).

HLA-F is a non-classical MHC class I molecule that interacts with NK cell receptors and is known to be upregulated in viral infections, including HIV. HLA-F-AS1, its antisense long non-coding RNA, may play a role in regulating HLA-F expression via epigenetic mechanisms. Although the role of HLA-F in CMV infection is less well defined compared to HLA-E, our findings suggest that epigenetic regulation in the HLA-F region may be involved in CMV-related immune responses, potentially through modulation of NK cell activity (DOI: [10.1038/ni.3513](https://doi.org/10.1038/ni.3513); DOI: [10.1002/eji.201040348](https://doi.org/10.1002/eji.201040348)).

Table R1. Summary statistics of the association of CpG sites within HLA-F and HLA-F-AS1 to CMV-seropositivity. This table has been included in the revised manuscript as Supplementary Table S14.

CpG	Gene	logFC	AveExpr	P.Value	adj.P.Val
cg08755130	HLA-F	-0.1724	7.4625	2.5361e-09	5.6940e-07
cg08755130	HLA-F-AS1	0.2469	2.4867	5.1418e-08	6.7101e-06

We have added the above the info to the **Results on page 15, lines 427-439**:

“Non-classical HLA molecules can modulate NK-cell checkpoints during CMV infection, especially HLA-E, which presents CMV-derived peptides to NK-cell receptor NKG2C⁵⁰. We therefore surveyed all omics layers for HLA-E signals. HLA-E mRNA in PBMCs was modestly higher in CMV+ individuals (P = 0.03, Supplementary Table S6), whereas circulating HLA-E protein and DNA methylation at the locus were unaltered. In the CMV GWAS, a sub-genome-wide signal lay adjacent to HLA-E (lead SNP rs2021368, $p = 4.25 \times 10^{-5}$; Supplementary Fig. S9F). By contrast, whole-blood methylation profiling identified a robust CMV-associated CpG at the HLA-F/HLA-F-AS1 locus (Supplementary Table S14; cg08755130, FDR = 5.7×10^{-7}), hinting that epigenetic regulation of another non-classical HLA gene that engages NK receptors contributes to CMV-driven immune modulation^{51,52}. Finally, we imputed classical HLA alleles with an HLA-specific reference panel to test whether MHC-I variation influences FCRL6 expression. We identified 4 HLA alleles that might affect the FCRL6 expression (P < 0.05); however, these alleles had not passed the multiple testing correction (Supplementary Table S15).“

Reviewer #3 (Remarks to the Author):

This is a comprehensive multi-omic analysis of the relationship of CMV seropositivity (latent CMV infection) with the immune system in people with HIV, as part of the 2000HIV project. The authors have observed profound effects of having CMV on gene expression, cytokine production, plasma proteome and metabolome, and DNA methylation in the peripheral blood and in peripheral blood cells, using a discovery and a validation cohort approach. While some of the differences are expected from previous literature, this study is notable for its very meticulous approach, the large numbers of individuals analyzed by each method, and the connections among the different assays, which demonstrate regulation at several levels - for instance, CMV is associated with DNA de-methylation of FCRL6 (consistent with its upregulated gene expression and protein abundance) in CMV-seropositive individuals with HIV. Finally, the authors use GWAS and mendelian randomization analysis to posit that the FCRL6 gene is causally linked to CMV infection, although how the authors think FCRL6 leads to CMV susceptibility is not well described.

We thank Reviewer #3 for acknowledging the thoroughness and scope of our multi-omics approach and for pointing out the need to better articulate the mechanistic role of FCRL6 in CMV susceptibility.

Major Concerns:

3.1. The data in Figure 1C, showing increased cytokines produced by PBMCs of CMV-seropositive donors after CMV pp65 stimulation for 24h seems to be a bit confusing to me. The authors don't really justify why CMV pp65 peptide (an internal matrix protein) should elicit innate cytokines directly. And an obvious alternative - that pp65 protein is processed and presented to CMV-reactive T cells in the culture, which then release cytokines (such as TNF or IFN γ) to activate the innate cells - is seemingly not considered. Indeed, this may explain the reduced TNF in the validation cohort after CMV pp65 stimulation (the darkest blue cell): innate cells are taking up the TNF.

The reviewer raises a very good point. The increase in innate cytokines in CMV-seropositive individuals upon pp65 stimulation is further confirming the presence of CMV-specific memory responses in PLHIV. It has been shown by others that CMV pp65, which is a late CMV antigen, triggers the response of CD8 $^+$ T cells, therefore increasing TNF and IFN-gamma production upon stimulation reflecting the presence of reminiscent CMV-specific CD8 $^+$ T cell responses, named as memory inflation (DOI: [10.1038/nri.2016.38](https://doi.org/10.1038/nri.2016.38)). Thus, although we did not measure IFN-gamma production upon CMV pp65 stimulation in the participants of our cohort, the cytokines TNF- α and IL-1 β production to CMV are proposed to originate from CMV-specific memory responses mediated by CD8 $^+$ T cells but also monocytes. It remains to be addressed whether monocytes can directly recognize CMV-derived peptide antigens. However, their activation is likely mediated by CMV pp65-specific CD8 $^+$ T cells, which recognize CMV pp65 peptides presented via MHC class I. For instance, IFN-gamma produced by these activated memory CD8 $^+$ T cells may, in turn, stimulate monocytes to produce cytokines (DOI: [10.1038/nri2448](https://doi.org/10.1038/nri2448)).

To indicate this more clearly to the reader we have reworked the **Results** section ("Exposure of PBMCs from CMV+ PLHIV to latent CMV antigen enhances the production of inflammatory

cytokines”) on **pages 7-8, lines 165-183** (renamed to: ‘Latent CMV infection heightens pro-inflammatory recall responses in PLHIV’):

“Among the 90 cytokine-stimulus read-outs measured, the most pronounced CMV-associated effects appeared after exposing PBMCs to the CMV antigen pp65 for 24 hours (Fig. 1C, Supplementary Tables S2-S3). In both the discovery and validation cohorts, CMV-seropositive (CMV+) donors secreted significantly higher amounts of the monocyte-derived cytokines IL-1 β , IL-1Ra, IL-8 and MCP1 than CMV-seronegative (CMV-) donors (FDR < 0.05 discovery; P < 0.05 validation) . This agrees with previous studies where heightened inflammatory responses in PBMCs to CMV infection for PLHIV is already well-documented¹¹⁻¹³.

As pp65 is an internal CMV protein that does not directly engage pattern-recognition receptors, the simplest explanation for the elevated innate-cytokine output is that the antigen is processed and presented to pre-existing pp65-specific T cells. Activated CD8+ T cells are known to release TNF-a and IFN- γ upon pp65 recognition, a phenomenon termed memory inflation, and the resulting IFN- γ allows monocytes to amplify IL-1 β , IL-1Ra, IL-8 and MCP-1 production^{14,15}. Consistent with this model, whole-blood immunophenotyping revealed higher frequencies of HLA-DR+ CD4+ and CD8+ T cells, NK cells and gamma-delta ($\gamma\delta$) T cells in CMV+ PLHIV (Fig. 1E; Supplementary Tables S4,S5). This mirrors the well-documented CMV-driven reshaping of CD4+, CD8+, NK-cell and $\gamma\delta$ -T-cell compartments in healthy adults^{15,16} and the capacity of CMV to upregulate HLA-DR expression on T cells¹⁷. Together, these findings establish that latent CMV infection primes a T-cell-driven, monocyte-dominated inflammatory response that distinguishes CMV+ from CMV- PLHIV.”

3.2. The authors go to a lot of effort to investigate the role of CMV on FCRL6 (and vice versa), but there is no data on which cells are expressing it in vivo, and whether cellular FCRL6 levels are different in CMV seropositive and seronegative individuals. They say from the literature that it is expressed on NK cells and T cells - could the authors stain PBMCs from CMV disparate individuals and show a difference in FCRL6 surface expression?

We thank the reviewer for this suggestion. To directly assess cellular FCRL6 expression, we performed flow cytometry on PBMCs from four PLHIV (2 CMV+, 2 CMV-) in our cohort. In short, cells were stained with markers for CD3, CD4, CD8, CD14, CD19, CD45, CD56, TCR $\gamma\delta$, and FCRL6. The resulting percentages of FCRL6+ cells within each subset can be found in Figure R-11, which indicate that CMV+ PLHIV have proportionally higher percentages of FCRL6 expressing CD8+ T cells as well as in total TCR $\gamma\delta$ T cells. In addition, CD14+ monocytes and NK cells (CD56+) expressing FCRL6+ were also higher in PLHIV CMV+ in comparison to PLHIV CMV-. Altogether, the current data provides additional evidence on the source of FCLR6 protein and further confirms its association with CMV seropositivity.

Figure R-11. Bar graph showing the percentage of FCRL6+ cells within the different immune cell subsets. The data is stratified based on the CMV status, including 2 PLHIV CMV+ and 2 PLHIV CMV-. This figure has been included in the revised manuscript as Supplementary Fig. S8B.

We added these new Results to the revised manuscript at page 13, lines 359-366:

“To pinpoint which immune cells contribute to the elevated plasma FCRL6 signal, we performed multicolour flow cytometry on PBMCs from four PLHIV in our cohort (two CMV+ and two CMV-). Cells were stained for CD3, CD4, CD8, CD14, CD19, CD45, CD56, TCR $\gamma\delta$ and FCRL6. CMV+ individuals showed higher proportions of FCRL6-expressing CD8+ T cells and $\gamma\delta$ T cells than CMV- counterparts (Supplementary Fig. S8). The frequency of FCRL6+ cells was also increased within CD14+ monocytes and CD56+ natural-killer cells. These data confirm that multiple cytotoxic and innate subsets up-regulate FCRL6 in vivo during CMV co-infection and provide a cellular source for the elevated protein we detect in plasma.”

And new Methods (‘Flow cytometry for FCRL6 in PLHIV PBMCs’, page 28, line 869-881):

“The PBMCs were cultured in RPMI 1640 medium (Gibco, Thermo Fisher Scientific, Waltham, MA, USA) supplemented with 10% human pool serum overnight and the stained for the following markers: LIVE/DEAD fixable viability dye viakrome (Beckman Coulter - Brea, CA, USA), CD3 (BV605 - Biolegend - San Diego, CA, USA), CD4 (BUV395 - Biolegend), CD8 (BV785 - Biolegend), CD14 (AF700 - Biolegend), CD19 (PE-Cy5 - Biolegend), CD45 (Krome Orange - Beckman Coulter - Brea, CA, USA), CD56 (BUV737 - Biolegend), TCR $\gamma\delta$ (BV421 - Biolegend), and FCRL6 (APC - Biolegend). Antibodies were diluted in a total staining volume of 50 μ L, which included Brilliant Stain Buffer Plus (BD Biosciences, Franklin Lakes, NJ, USA) and PBS supplemented with 2% bovine serum albumin (BSA-Sigma Aldrich, USA) and 2 mM EDTA (Gibco) (PBS-F) as the staining/washing buffer. For FcRL6-APC, a fluorescence minus one (FMO) control was included to define the positive populations. Staining was performed for 30 minutes at 4°C in the dark, followed by washing with PBS-F + EDTA. Data acquisition was carried out using the CytoFLEX flow cytometer (Beckman Coulter, Brea, CA, USA) (Supplementary Fig. S8A).”

3.3. Could the authors discuss why none of the SNPs (instrumental variables) shown in Figure 5C (causal for gene expression) overlap with the SNPs in Figure 5E (causal for protein expression), and vice versa. Is this an expected finding or not?

We thank the reviewer for their insightful question. Indeed, the instrumental variables (IV) for FCRL6 gene expression and protein abundance do not overlap, indicating that the genetic regulation of FCRL6 at the transcript and protein levels appears to be independent in this context. This is an expected finding and aligns with previous studies (DOI: [10.1038/s41586-018-0175-2](https://doi.org/10.1038/s41586-018-0175-2)), showing that eQTLs and pQTLs can be driven by different genetic variants due to post-transcriptional regulation, differences in mRNA and protein stability, etc.

Additionally, the differences in genetic regulation could also be explained by the fact that we measured gene expression in PBMCs and protein levels in plasma. PBMC gene expression primarily captures intracellular transcriptional regulation, whereas plasma protein levels reflect systemic circulation.

Lastly, most of the IV are trans-QTLs, where the genetics variant affects the gene or protein at a different genomic locus. Given that trans-QTLs generally have smaller effect sizes compared to cis-QTLs, statistical power is more limited. Despite this, we identified two SNPs within the same genomic region that, although not in linkage disequilibrium (LD), suggest the possibility of shared regulatory mechanisms.

Importantly, both the gene and protein level MR results for FCRL6 show significant associations with CMV serostatus, and the overall conclusion is supported by converging lines of evidence rather than a single SNP or QTL.

We have added some text to discuss this matter to the revised manuscript (Discussion, **page 15, lines 415-423**):

“Notably, we observed no overlap between the IVs for FCRL6 gene expression in PBMCs (Fig. 5C) and those for protein abundance in whole blood (Fig. 5E). This is consistent with previous studies showing that eQTLs and pQTLs commonly involve distinct genetic variants, reflecting differences in post-transcriptional and translational regulation⁴⁷. However, we did identify two SNPs located in close proximity on chromosome 14 (rs140233210 at position 20,061,131 for gene expression and rs1959651 at position 20,080,570 for protein expression). Although these SNPs are not in linkage disequilibrium (LD), their close genomic proximity suggests that some shared regulatory mechanisms influencing FCRL6 expression at different biological levels might exist, warranting further investigation.”

Minor Concerns:

- Please refrain from using terms like “HIV-infected” or “non-HIV-infected”. Just use people (living) with HIV or people without HIV. The term “patients” is also discouraged.

Thank you for your feedback. We have revised the manuscript to replace terms like “HIV-infected” and “non-HIV-infected” with “people living with HIV” and “people without HIV” throughout the text.

Additionally, we have deleted the word “patients”, e.g. replacing “transplant patient” with “transplant recipient”.

- Table S7 only seems to contain 700 DEGs (FDR <0.05), not 807 as stated in the text (line 257).

Thank you for this comment. The validation was originally performed using the nominal p-value, resulting in 807 DEGs. During this revision, we apply the FDR < 0.05 in the validation cohort, resulting in 700 DEGs as mentioned in the comment. We updated this information in the manuscript's main text.

- Figure 5 panel E is not labeled within Figure 5.

Thank you for catching this. We have now labeled in the revised manuscript.

- Line 587 in the Methods says “(show figures)” ???

We apologize for this mistake. The entire sentence this is referring to has been removed from the Methods section.

- Figure 2B legend (lines 769-771) the x-axis and y-axis designations are reversed from what is shown in Figure 2B.

Thank you for pointing this out. We corrected the mistake in the manuscript.

- Figure 2F legend (line 780) says “MOMER” instead of “HOMER”.

Thank you for pointing this out. We corrected the mistake in the manuscript.

Reviewer #4 (Remarks to the Author):

This is an interesting manuscript that assesses CMV serostatus in relation to a large study of 'omics data in a cohort of people living with HIV. The work develops hypotheses related to the impact of CMV on the immune system.

We are grateful to Reviewer #4 for your interest in our work examining CMV serostatus within a large HIV-infected cohort and for noting the hypothesis-generating strength of our omics analyses.

Points

4.1. The clinical cohort has a very high rate of CMV seroprevalence and this may reflect demographic factors related to infection risk. The relevance of the data to the general population is thus not clear and this should be stated as a limitation.

Thank you for your insightful comment regarding the high rate of CMV seroprevalence in our clinical cohort. We agree that the unusually high CMV seroprevalence (~94 %) in our cohort likely reflects its demographic make-up (predominantly middle-aged MSM living with HIV in the Netherlands) and could limit external generalisability. To address this, we have added a paragraph to the **Discussion (page 18, lines 523-533)** that (i) contrasts our prevalence with national estimates, (ii) explains how high baseline exposure might affect effect-size estimates, and (iii) calls for replication in cohorts with lower CMV prevalence. Despite this, the high replication in our down-sampling analysis, as well as the consistent signal of FCRL6 across all omics layers, lend confidence to the biological relevance of our findings despite the epidemiological limitation.

“CMV seroprevalence in the 2000HIV study is ~94%, which is considerably higher than the ~50% reported for age-matched adults in the general Dutch population⁶⁶. This elevation almost certainly reflects the cohort’s demographic composition, which is predominantly middle-aged men who have sex with men (MSM) living with HIV and using long-term ART. High baseline prevalence reduces statistical power to study CMV- PLHIV and may exaggerate the magnitude of CMV-associated phenotypes. However, the downsampling analyses showed high concordance with the full-cohort results, with the vast majority of significant associations replicated across iterations, demonstrating robust reproducibility. Even so, the cytokine and multi-omic signatures reported here should be interpreted with caution in settings where CMV exposure is less common, and future multi-centre studies that include low-prevalence populations will be required for external validation.”

4.2. I find the manuscript to be rather poorly written for the scientific reader. I would advise a considerable rewriting that addresses the cohort structure, experiments that were done in each section and the specific outcomes. The abstract also needs to be succinct and clear

We thank the reviewer for this writing style suggestion. The 2000HIV study has been previously published by our collaborators (DOI: [10.3389/fimmu.2022.982746](https://doi.org/10.3389/fimmu.2022.982746)), including study design, multi-omics methods, and participant characteristics information. Since our manuscript focuses on bioinformatic analysis, we emphasized the data analysis steps, and only presented the key

information about cohort structure and wet-lab experiments while citing the original paper. In response to your suggestion, as well as the other reviewer's suggestions, we have rewritten the abstract and the main text to be more concrete and scientific-oriented.

4.3. The cytokine functional assessment is in relation to the short term response to challenge with a CMV protein. CMV modulates the immune systems in substantial ways and as such this differential response is to be expected.

We agree with the reviewer that this is expected. However, we also feel the present manuscript aims to provide further insights on the effects of CMV reactivation in the context of people living with HIV. For instance, as mentioned above to Reviewer #3, the short-term increase in pro-inflammatory cytokine production in response to CMV pp65 could be due to specific memory responses. These CMV responses may predispose PLHIV to poor prognosis as it has been suggested that CMV-driven inflammation can lead to increased HIV replication and reservoir (DOI: [10.1093/infdis/jiw217](https://doi.org/10.1093/infdis/jiw217); [10.1089/aid.2017.0145](https://doi.org/10.1089/aid.2017.0145)). Therefore, a deeper understanding of the effects of CMV provides opportunities to identify therapeutic targets that reduce the burden of CMV-co-infections in PLHIV.

4.4. The work uses bulk transcriptomics and the increase in many cytotoxic transcripts is not surprising.

We agree that this pattern is not unexpected given that the data is from PLHIV. We feel our ability to recapitulate them using independent omics measurements highlights both the quality of our dataset and the robustness of our analytical approach.

4.5. CMV induces major changes in the immune cell repertoire - has this been corrected for in relation to the relative prevalence of genes and methylation profiles associated with this transition?

We appreciate the reviewer's comment regarding immune cell composition changes induced by CMV infection and their potential impact on methylation and gene expression profiles. Although we do not have direct measurement of the immune cell repertoire, we have corrected for the confounding effects for each omics layer, as mentioned in the Methods section.

For the **methylation** profile, to account for differences in immune cell proportions, we estimated the relative abundance of six major immune cell types (neutrophils, monocytes, B cells, NK cells, CD8+ T cells, and CD4+ T cells) using DNA methylation data. Specifically, we applied a modified version of Houseman's method implemented in the `estimateCellCounts2` function from the `FlowSorted.Blood.EPIC` R package. These estimated cell-type proportions were included as covariates in downstream methylation analyses to correct for potential confounding due to shifts in immune cell composition associated with CMV seropositivity.

For **gene expression** profiles, we adjusted for the top confounding effect of age, sex, BMI, center collection, season (`season_sin`, `season_cos`), and ethnicity (top5 genetic PCs) in the discovery cohort.

Since the validation cohort had a smaller sample size and was collected in one center site, we adjusted for the confounding effect of age and season (season_sin, season_cos).

4.6. The work also seeks to assess genomic risk factors related to CMV infection. There has been prior work in this regard and the incremental advance here seems modest. The potentially unusual features of the residual CMV-seronegative donors within this cohort should be discussed.

We agree that a comprehensive discussion on the unusual features of the residual CMV-seronegative will benefit the related field. To our knowledge, there are 21 reported genetic loci associated with CMV infection in different disease contexts at a p-value threshold of $5e-5$, such as patients after allogeneic hematopoietic cell transplantation and patients who are diagnosed Schizophrenia. These loci are from three cohorts consisting of European participants. These studies are mainly focused on specific disease populations, which is just like our scenario. In our study, we observed ultra high CMV-seropositive rates in people living with HIV, which suggests a high chance of co-infection of HIV and CMV. In contrast, a hand of PLHIV is CMV-seronegative. It indicates protective factors in this small group of donors and the underlying protective factors are extremely important to help patients' resistance to CMV infection. Of note, the above mentioned studies have indicated genetic loci potentially involved in the CMV infection, which encouraged us to explore the potential genetic factors resisting CMV infection regardless of the high risk. The GWAS results suggested a genetic locus, which supported our hypothesis and could be concluded that genetic factors may play an important role in resistance to CMV infection in people living with HIV.

Following the reviewer's suggestion, we have added the following text to the revised manuscript (**pages 17-18, lines 511-522**) to address this:

“Previous studies have suggested that genetic factors are potentially involved in the CMV infection in different disease contexts, such as individuals after allogeneic hematopoietic cell transplantation and individuals who are diagnosed Schizophrenia⁶³⁻⁶⁵. However, the potential roles of genetic variants in resistance to CMV infection in PLHIV is less known, in contrast to the observed high prevalence of CMV seropositivity. Therefore, we performed a GWAS to screen genetic loci that are associated with CMV- in our PLHIV cohort. The screening suggests a locus on chromosome 15 where rs7180928-G allele is protective, which is not previously reported. To confirm its role in CMV infection in our study cohort, we systematically estimated whether this locus bridges other molecular traits with CMV-sero-negativities in PLHIV. The analysis suggested interplays among the identified locus, cytokine production, protein abundance, and gene expression, which collectively highlighted the importance of the identified locus in CMV resistance upon HIV-positive.”

4.7. The FCRL6 association is interesting and the most important feature of the manuscript. The magnitude of this effect, and as such its potential use in assessment of CMV status within individual donors, could be expanded.

We thank the reviewer for recognising FCRL6 as a key finding in our study. As shown in Table R2, multiple CpG sites within the FCRL6 locus exhibit strong and consistent hypomethylation in

CMV-seropositive individuals (e.g., cg24833981: logFC = -0.7956, FDR = 4.9×10^{-18}), and our eQTM analysis further links this epigenetic signature to increased FCRL6 expression. To display the strength of the epigenetic effect more clearly, we have now included this table in our revised manuscript, with a reference to it on **page 15, line 434**. Importantly, we also observe elevated FCRL6 protein levels in CMV+ donors (Figure 3), underscoring its multi-layer biomarker potential. While the downsampling analysis confirms that these associations are robust to the imbalance in CMV status, we acknowledge that the limited number of CMV- individuals in the 2000HIV cohort constrains definitive conclusions about clinical assay performance. Therefore, we have moderated our claims regarding FCRL6's immediate diagnostic utility and propose that future work that leverages larger, more balanced cohorts, will be essential to fully validate FCRL6 as a reliable marker of CMV infection.

Table R2. Summary statistics of the association of CpG sites within *FCRL6* to CMV-seropositivity. *This table has been included in our revised manuscript as Supplementary Table S14.*

CpG	Gene	logFC	AveExpr	P.Value	adj.P.Val
cg23615393	FCRL6	-0.4498	5.1535	1.4797e-06	0.0001236
cg24833981	FCRL6	-0.7956	5.1535	1.0268e-21	4.8908e-18
cg00950718	FCRL6	-0.2619	5.1535	3.4875e-05	0.0006598
cg05919685	FCRL6	-0.3901	5.1535	1.2677e-06	6.0838e-05
cg27198652	FCRL6	-0.5346	5.1535	9.7049e-10	1.0750e-07

Reviewer #4 (Remarks on code availability):

NA

Reviewer #1 (Remarks to the Author):

I appreciate the authors' efforts during the revision. The manuscript has been improved significantly. Here are a few points I hope could be elaborated further.

We thank Reviewer #1 for noticing our efforts and finding further things that would improve the manuscript.

1)Major point 1.

I appreciate the downsampling analysis and supFig S3,S4,S5. Just out of curiosity, how was the number of downsample size '86' determined? Any intuition behind and maybe included in the methods section?

We thank the reviewer for this question. As mentioned in the previous response letter, there were 86 CMV- individuals in the discovery cohort, therefore we matched this with 86 CMV+ individuals for the downsampling analysis. This is also mentioned in the methods 'Downsampling analysis' on page 26. However, we see now that we don't explicitly mention the specific number in the result section of the manuscript, just the percentages. We have now made additions (bolded below) to the result section that describes the cohort to mention the sample sizes of the CMV- individuals within the discovery and validation cohorts to the manuscript on page 7 line 150:

"Baseline cytomegalovirus (CMV) IgG was measured in all 1,887 participants by ELISA, revealing seronegativity in ~6% of the discovery cohort (**n = 86**) and ~8% of the validation cohort (**n = 28**) (Supplementary Table S1)."

A minor suggestion for FigS4-S5: Maybe the FCRL6 can be highlighted using a different color or shape for the point. Also the 'validate_DEGs/DEPs yes/no' labels only need to show once for the whole plot instead of in each subplots.

We thank the reviewer's suggestion for improving the visualisation of Figures S4-5. The FCRL6 were highlighted in a different color (dark red color). The labels are grouped and shown once at the bottom for the whole plot, including for S3.

2)Major point 4.

2.1 Could the authors provide PPH1 and PPH2 of the coloc analysis as well?

We thank the reviewer's suggestion and think that supplying PPH1 and PPH2 helps improve the manuscript. Following the request, we have updated the manuscript by adjusting the following sentence to include the PPO, PP1, and PP2 from the colocalization analysis.

"To further confirm the role of CHRN4 in CMV infection in PLHIV, we performed colocalization and MR analyses to verify if the gene mediates PLHIV's resistance to CMV. The colocalization

resulted in a PP4 = 0.34, PP3 = 0.23, PP2 = 0.39, PP1 = 0.02, and PP0 = 0.03 (Bayesian factor analysis), ...”.

2.2 Could the author also add a box plot for CHRNA4 eQTL? Please also report the accurate nominal p value of the CHRNA4 eQTL for rs7180928 instead of ‘P<0.05’.

Following the reviewer’s suggestion, we have added the following violin plot to display the eQTL of CHRNA4 as Supplementary figure S9D. We also updated the manuscript to mention the newly added supplementary figure. The exact p-value (p-value = 2.13e-05) has also been updated in the manuscript (at line 383 of page 14).

Figure R-2.2. Violin plot showing the eQTL effect of rs7180928 for gene CHRNA4. The observed p-value is 2.13e-05. Included in the revised manuscript as Supplementary Fig. S9D.

2.3 I appreciate the new locus zoom plot with LD information. But the Figure legend for Fig4C (line) seems to be not updated (there is no orange dots anymore). Now the eQTL and GWAS signals are not easy to distinguish (dark blue vs light blue?). Could the authors try to improve it? Maybe another more straightforward way is to plot them separately but just stack the GWAS on top of the eQTL and use the same gene annotation panel. It would be better to have 2 y axis (left and right) showing for both -log10p value of eQTL and of GWAS, if you still overlay the 2 plots.

We apologise for not updating the figure legend previously. Following the reviewer’s suggestion, we have updated the figure by stacking two Manhattan plots but keeping the original gene panel, which is available as the following. And the figure legend has been updated as well.

Figure R-2.3. Updated locuszoom plot which stacks GWAS and eQTL summary statistics of locus pinpointed by rs7180928. Included in the revised manuscript as Fig. 4C.

2.4 According to the original Figure4C, the lead eQTL for CHRN4 is rs11629784. What is the p value of this one in the GWAS?

We thank the reviewer's question and also think the p-value of rs11629784 in CMV status helps to increase the manuscript. However, the rs11629784 is excluded in the GWAS analysis due to low frequency (MAF = 0.006963) in the discovery cohort and there is no donor that is grouped into AA genotype. As stated in our method section, we ensure 3 donors in each genotype group to guarantee the logistic regression works smoothly, but we release this restriction in eQTL analysis when using linear regression. We added the following sentence to underscore the limitation at line 383 on page 14.

"However, the top cis-eQTL SNP of CHRN4 is rs11629784, but this SNP is excluded in the GWAS analysis due to low frequency (MAF = 6.96e-3) in the discovery cohort."

2.5 I appreciate the author's effort of finding CHRN4 related studies. However, I am still not convinced this is the most likely mediating gene. According to GTEX data, rs7180928 has eQTL effect for CRABP1(p=0.0000041) and for DNAJA4(p= 0.000016) (see <https://gtexportal.org/home/snp/rs7180928>), are they not significant or not detectable in your analysis?

We respect the reviewer's comment. As indicated in our method section, transcriptional profiles are from PBMCs isolated from the donors' whole blood, while the reported eQTLs of CRABP1 and DNAJA4 at rs7180928 by GTEx are from thyroid and nerve (tibial), respectively. Moreover, the rest reported eQTL genes by GTEx for this given SNP are not from whole blood either. Therefore, we did not report these eQTLs in our last manuscript. Of note, their expressions are low in whole blood which limits the power to detect eQTL, for instance CRABP1 (<https://gtexportal.org/home/gene/ENSG00000166426.8>).

2.6 There are quite a bunch of genes (e.g. SKIC8, DNAJA4, CRABP1, IREB2, HYKK, PSMA4, CHRNA3, CHRNA5) near the GWAS peak in this locus. I assume the authors have full eQTL summary statistics for all these genes. I guess some of these genes in addition to CHRN4 should also have eQTL signal. I would suggest the authors to run coloc between each of these eQTLs and GWAS, then to check if the CHRN4 eQTL really has the highest PPH4. This analysis should be able to make sure that CHRN4 is the most likely mediating gene.

We appreciate the reviewer's suggestion to present colocalization analyses for genes in the identified locus. To improve the manuscript, we added a supplementary table (Supplementary Table S16) to release the concerns. The following table (Table R1) shows the genes suggested by the reviewer.

Table R1. Small snippet of new Supplementary Table S16 showing the genes suggested by the reviewer.

Feature	N SNPs	PPH0	PPH1	PPH2	PPH3	PPH4	Comment
CHRN4	579	2.61E-02	1.53E-02	0.39131	0.22945	0.337789	Reported gene by this study
CRABP1	579	5.65E-02	3.32E-02	0.52821	0.31010	0.071985	
DNAJA4	579	1.81E-18	1.06E-18	0.59931	0.35188	0.048815	
PSMA4	579	8.03E-06	4.72E-06	0.59962	0.35206	0.048309	
WDR61	579	1.36E-12	7.98E-13	0.60097	0.35286	0.046171	a.k.a. SKIC8
CHRNA3	579	2.34E-15	1.37E-15	0.60294	0.35401	0.043050	
CHRNA5	579	6.36E-26	3.73E-26	0.60345	0.35432	0.042235	
IREB2	579	6.43E-49	3.77E-49	0.60402	0.35465	0.041332	
HYKK	579	1.54E-01	9.04E-02	0.45188	0.26531	0.038570	

3)Major point 5.

Please add the KIR2DS4 and IL22 to the y axis labels of the Fig4F,G. Also for this kind of QTL box plots, I think the authors should have a p value for each, instead of comparing different genotypes 3 times. Could you label that p value from the QTL analysis? Please also report the test method in the figure legend.

We thank the reviewer's suggestion which surely helps to improve the manuscript. We have updated the figures by adding "KIR2DS4" and "IL22" to the y-axis labels. The figures are also updated by adding exact p-values of Spearman correlation between genetic dosage and protein abundance or cytokine level, which is indicated in the figure and figure legends. The exact p-values of Fig 4F and 4G are also updated in the manuscript. For Fig 4F, $r = -0.11$, $p = 1.3e-4$, Spearman correlation test (one-way, less than 0) and for Fig 4G, $r = -0.07$, $p = 2.7e-2$, Spearman correlation test (one-way, less than 0), respectively, which have been updated in the figures, which are also available as the following.

Figure R-3. The updated violin plots by adding statistical tests and updating labels of y-axis. Included in the revised manuscript as Fig 4F.

All the other points were well addressed, thanks.

We thank the reviewer for this comment.

Please also update the code for the new analysis on github.

We have now updated the code for the new analyses from the previous revision and this revision on GitHub at https://github.com/CiiM-Bioinformatics-group/2000HIV_CMV.

Reviewer #2 (Remarks to the Author):

I am satisfied with the changes the authors have made

Reviewer #2 (Remarks on code availability):

The code has been checked and was functional and appropriate

We thank Reviewer #2 for these comments.

Reviewer #3 (Remarks to the Author):

The authors have done an outstanding job responding in great detail to the reviewers' comments, including my own. The manuscript is much improved and I don't have any major concerns that would preclude acceptance.

Minor concerns:

-There are some typos (words without spaces between, etc) and other English language issues, particular on lines:

52 ("populationsassociated")

52 ("seropositivity.This")

57 ("PLHIV. ost")

103 ("to90%")

240 ("cytokines PBMCs")

294-295 ("also occur HIV-specific")

320 ("not unsurprising") [I think you mean not surprising here]

-In addition, many of the references appear to be messed up, with only initials given instead of names: Refs 4, 6, 14, 16, 17, 19, 20, 21, 22, 24, 25, 26, 27, 28, 30, 31, 32, 34, 35, 36, 43, 44, 46, 48, 49, 53, 54, 55, 57, 58, and 72.

We thank Reviewer #3 for these comments, as well as noticing these typos and reference issues. It appears we created our references database exactly during the weekend when there was a website maintenance ongoing at NCBI back in July. We have now refetched the metadata for these references and the issue has been resolved.

Reviewer #4 (Remarks to the Author):

This is a revision.

The authors have answered the reviewers comments very well.

They have maximised the power of the study cohort.

The demonstration of increased levels of FCRL6 on several immune cells is important.

There are spelling mistakes in the abstract which is surprising.

We thank Reviewer #4 for these comments. We also apologise for the spelling mistakes in the abstract. We hope we have now found and fixed all of these.